# Towards Better Understanding Open-set Noise in Learning with Noisy Labels

## Abstract

To reduce reliance on labeled data, learning with noisy labels (LNL) has garnered increasing attention. However, most existing works primarily assume that noisy datasets are dominated by closed-set noise, where the true labels of noisy samples come from another known category, thereby overlooking the widespread presence of open-set noise—where the true labels may not belong to any known category. In this paper, we refine the LNL problem by explicitly accounting for the presence of open-set noise. We theoretically analyze and compare the impacts of open-set and closed-set noise, as well as the differences between various open-set noise modes. Additionally, we examine a common open-set noise detection mechanism based on prediction entropy. To empirically validate our theoretical insights, we construct two open-set noisy datasets—CIFAR100-O and ImageNet-O—and introduce a novel open-set test set for the widely used real-world noisy dataset, WebVision. Our findings indicate that open-set noise exhibits distinct qualitative and quantitative characteristics, underscoring the need for further exploration into how models can be fairly and comprehensively evaluated under such conditions.

## 1 Introduction

In recent years, the remarkable success of machine learning has largely relied on the assumption that data labels are accurate and noise-free. However, in real-world scenarios, label noise—arising from factors such as annotation errors and label ambiguity—is pervasive, posing significant challenges to model performance and generalization. To address this issue, various approaches have been proposed for learning with noisy labels (LNL), including noise transition matrix estimation (Goldberger and Ben-Reuven, 2017; Xia et al., 2019; 2022), noisy label correction (Song et al., 2019; Cascante-Bonilla et al., 2021), robust loss functions (Ghosh et al., 2017; Zhang and Sabuncu, 2018; Wang et al., 2019), and, more recently, dominant sample selection-based methods (Han et al., 2018; Arazo et al., 2019; Li et al., 2020; Xia et al., 2021; Feng et al., 2022).

Most current efforts primarily focus on closed-set noise, where the true labels of noisy samples belong to another known class. This includes common noise models such as symmetric noise, where sample labels are randomly flipped to any other known class with a certain probability, and asymmetric noise, where label confusion is influenced by class similarity (e.g., 'cat' is more likely to be confused with 'dog' than with 'airplane'). Recent advances have also explored instance-dependent noise models (Chen et al., 2021; Yang et al., 2022), where label confusion is directly influenced by the semantics of individual instances.

However, unlike the extensive research on closed-set noise, there is significantly less focus on open-set noise, where the true labels of noisy samples do not belong to any known category. This gap is particularly noteworthy given that one of the primary motivations for learning with noisy labels is to manage datasets collected through web crawling. By examining one of the most commonly used benchmarks, the WebVision dataset (Li et al., 2017), we confirm the prevalence of open-set noise (fig. 1).

In fact, the "open-world" assumption, which involves open-set samples, has received considerable attention in other weakly supervised learning problems, such as open-set recognition and outlier detection. However, it remains underexplored in the context of learning with noisy labels. To address this gap, this paper focuses on a comprehensive theoretical analysis of open-set noise. The main findings are outlined as follows:

Figure 1: Example images of class "Tench" from WebVision dataset - Clean samples are marked in green, closed-set noise is marked in blue, and open-set noise is marked in red. See appendix H for more discussions.

- We introduce the concept of a complete noise transition matrix, reformulate the Learning with Noisy Labels (LNL) problem to account for open-set noise, and analyze two offline cases: *fitted* and *overfitted*.
- We demonstrate that open-set noise generally has less negative impact on classification accuracy than closed-set noise, analyze 'hard' vs. 'easy' open-set noise, propose an out-of-distribution (OOD) detection task for further evaluation, and find entropy-based open-set noise detection effective for 'easy' open-set noise.
- We conduct preliminary explorations with vision-language models and self-supervised models on identifying and learning with 'hard' open-set noise, expand experiments on the performance of robust loss functions under open-set noise, and analyze their effectiveness in challenging noise scenarios.

## 2 RELATED WORKS

In this section, we provide a brief overview of mainstream LNL methods, relevant research connected to this work, and the key motivations derived from them. Briefly speaking, methods for learning with noisy labels can be roughly categorized into two main types.

**Statistical-consistent methods**   The first type is often referred to as statistical-consistent methods, such as estimating noise transition matrix (Chen et al., 2021; Yang et al., 2022; Xia et al., 2019; Goldberger and Ben-Reuven, 2017; Liu et al., 2023; Wang et al., 2024) or designing robust loss functions (Zhang and Sabuncu, 2018; Wang et al., 2019; Ghosh et al., 2017; Chen et al., 2023; Mao et al., 2023; Patel and Sastry, 2023a; Wilton and Ye, 2024), aiming to achieve theoretically risk-consistent or probabilistically-consistent models. However, most of these works often assume an ideal scenario where the model can learn to fit the sampled distribution well, overlooking the over-fitting issues arising from excessive model capacity and insufficient data in practical situations. In this paper, we introduce the concept of complete noise transition matrix that accounts for open-set noise and conduct theoretical analyses and experimental validations for both ideal case and over-fitting case, namely the ***fitted case*** and ***overfitted case***.

**Statistical-approximate methods**   The second type, often referred to as statistical-approximate methods, includes dominant sample selection-based approaches that incorporate various regularization terms and off-the-shelf techniques such as semi-supervised learning and model co-training to achieve state-of-the-art performance. Most sample selection methods rely on the model's current predictions, such as the popular 'small loss' mechanism (Arazo et al., 2019; Li et al., 2020; Han et al., 2018; Yu et al., 2019; Jiang et al., 2018), and various improved variants upon it (Song et al., 2019; Malach and Shalev-Shwartz, 2017; Yi and Wu, 2019; Xia et al., 2021; Zhou et al., 2020; Wang et al., 2022; Cordeiro et al., 2021b; Patel and Sastry, 2023b). Some other works attempt to use feature representations for sample selection. Wu et al. (2020) and Wu et al. (2021) try to construct a graph and identify clean samples through connected subgraphs, while Feng et al. (2022) and Ortego et al. (2021) suggest directly using kNN in feature space to mitigate the impact of noisy labels. Moreover, as hybrid methods, sample selection approaches often involve additional auxiliary techniques, such as model co-training (Han et al., 2018; Yu et al., 2019; Wei et al., 2020; Zhao et al., 2024; Sun et al., 2021; Cordeiro et al., 2021a), semi-supervised learning (Li et al., 2020; Arazo et al., 2019), and contrastive learning (Li et al., 2022; Ortego et al., 2021; Huang et al., 2023; Zheltonozhskii et al., 2021; Ghosh and Lan, 2021; Karim et al., 2022).

**Exploration of open-set noise**  Research on open-set noise remains relatively limited. Wang et al. (2018) use the Local Outlier Factor algorithm to detect open-set noise in the feature space, while Wu et al. (2021) propose identifying open-set noise through subgraph connectivity. Both Sachdeva et al. (2021) and Albert et al. (2022) focus on entropy-related dynamics to identify open-set noise. In contrast, Feng et al. (2022) avoid explicitly identifying open-set noise and instead prevent relabeling or including it in the training process. More closely related to our work, Xia et al. (2022) also investigate noise transition matrices that account for open-set noise but assume all open-set noise belongs to a single meta-class. In this paper, we extend this idea by considering that open-set noise may originate from multiple classes. Based on this premise, we analyze two distinct modes of open-set noise. Wei et al. (2021) suggest leveraging open-set noise to mitigate the impact of closed-set noise, as it helps reduce overfitting. However, our focus is on providing a thorough theoretical analysis of the effects of different noise modes, including open-set noise versus closed-set noise, as well as comparisons between different types of open-set noise.

## 3 METHODOLOGY

In section 3.1, we briefly introduce the problem formulation of LNL and extend it to account for open-set noise. In section 3.2, we formalize how label noise affects model generalization, particularly focusing on the proposed error rate inflation metric. In section 3.3, we analyze and compare the impact of open-set *vs.* closed-set noise, as well as 'easy' open-set noise *vs.* 'hard' open-set noise. In section 3.4, we scrutinize the open-set noise detection mechanism based on model prediction entropy values.

### 3.1 REVISITING LNL CONSIDERING OPEN-SET NOISE

Supervised classification learning typically assumes that we sample a certain number of independently and identically distributed training samples $\{\boldsymbol{x}_k, y_k\}_{k=1}^K$ from a joint distribution $P(\mathbf{x}, \mathrm{y}; \mathrm{y} \in \mathcal{Y}^{in})$, i.e., the so-called training set. By default, here all the possible values for $y_k$ in the discrete label space $\mathcal{Y}^{in} : \{1, 2, ..., A\}$ (referred here as *inlier classes*), are known in advance. With a certain loss function, given the training set $\{\boldsymbol{x}_k, y_k\}_{k=1}^K$ we aim to train a model $f : \boldsymbol{x} \to y$ whose predictions can achieve the minimum classification error rate over the whole joint distribution $P(\mathbf{x}, \mathrm{y}; \mathrm{y} \in \mathcal{Y}^{in})$. Under LNL problem setting, we assume that the conditional distribution $P(\mathrm{y}|\mathbf{x}; \mathrm{y} \in \mathcal{Y}^{in})$ has been perturbed to $P^n(\mathrm{y}|\mathbf{x}; \mathrm{y} \in \mathcal{Y}^{in})$, leading to the presence of noisy labels $y_k^n$ in the noisy training set $\{\boldsymbol{x}_k, y_k^n\}_{k=1}^K$ that do not conform to the clean conditional distribution $P(\mathrm{y}|\mathbf{x}; \mathrm{y} \in \mathcal{Y}^{in})$.

In this work, instead of assuming all the possible classes are known ($\mathrm{y} \in \mathcal{Y}^{in}$), we consider samples from unknown outlier classes may also exist in the training set. Let us denote these classes as *outlier classes* $\mathcal{Y}^{out} : \{A+1, A+2, ..., A+B\}$, where $B$ represents the number of outlier classes[1]. Then, we expand the support of joint distribution to contain both inlier and outlier classes, denoted as $P(\mathrm{y}|\mathbf{x}; \mathrm{y} \in \mathcal{Y}^{in} \cup \mathcal{Y}^{out})$ and $P^n(\mathrm{y}|\mathbf{x}; \mathrm{y} \in \mathcal{Y}^{in} \cup \mathcal{Y}^{out})$ for the clean and noisy ones, respectively. For brevity, we denote the combined label space as $\mathcal{Y}^{all} \triangleq \mathcal{Y}^{in} \cup \mathcal{Y}^{out}$. For subsequent analysis, we first define below complete noise transition matrix:

**Definition 3.1** (Complete noise transition matrix). For a specific sample $\boldsymbol{x}$ (sample index omitted here for simplicity), we define as $T$ the complete noise transition matrix

$$T = \{T_{ij}\}_{i,j=1}^{A+B} = \left[ \begin{array}{c|c} T_{in_{A \times A}} & \mathbf{0}_{A \times B} \\ \hline T_{out_{B \times A}} & \mathbf{0}_{B \times B} \end{array} \right]_{(A+B) \times (A+B)}$$

Here, we denote as $T_{ij} \triangleq P(\mathrm{y}^n = j | \mathrm{y} = i, \mathbf{x} = \boldsymbol{x}; \mathrm{y}^n, \mathrm{y} \in \mathcal{Y}^{all})$. Note that, unlike existing literature, we do not require the noise transition matrix to be class-dependent. The matrix $T_{in}$ represents the confusion process between different inlier classes $\mathcal{Y}^{in}$, and $T_{out}$ captures the confusion process from outlier classes $\mathcal{Y}^{out}$ to inlier classes $\mathcal{Y}^{in}$. We explicitly define as $T^{out}$ the *open-set noise mode*. The right-hand side (highlighted in gray) contains all-zero entries, as we assume in the noisy labelling process the outlier classes are agnostic(unknown), i.e., all of collected samples will be labelled as one of the inlier classes.

---

[1] The subsequent analysis is independent of the specific values of $A$ and $B$, although it is generally expected that $B > A$.

For a specific sample $\boldsymbol{x}$ with such a complete noise transition matrix $T$, we can relate its clean conditional distribution $P(\mathrm{y}|\mathbf{x} = \boldsymbol{x}; \mathrm{y} \in \mathcal{Y}^{all})$ with its noisy conditional distribution $P^n(\mathrm{y}|\mathbf{x} = \boldsymbol{x}; \mathrm{y} \in \mathcal{Y}^{all})$ as follows:

$$P^n(\mathrm{y} = j|\mathbf{x} = \boldsymbol{x}; \mathrm{y} \in \mathcal{Y}^{all}) = \sum_{i=1}^{A+B} P(\mathrm{y} = i|\mathbf{x} = \boldsymbol{x}; \mathrm{y} \in \mathcal{Y}^{all}) \cdot T_{ij} \tag{1}$$

**Label noise** Recent works usually discriminate label noise into closed-set noise and open-set noise. For example, most recent studies define open-set noise as 'a sample with its true label from unknown outlier classes but mislabelled with a known label from inlier classes'. Before continuing with the further discussion, it is necessary to clearly define these two concepts here clearly to avoid any ambiguities, as we will try to comparably discriminate and analyze them later. Formally, we have:

**Definition 3.2** (Label noise). For a sample $\boldsymbol{x}$ with clean label $y$ and noisy label $y^n$:

- When $y = y^n$, $(\boldsymbol{x}, y, y^n)$ is a clean sample;

- When $y \neq y^n$ and $y \in \mathcal{Y}^{in}$, $(\boldsymbol{x}, y, y^n)$ is a closed-set noise;

- When $y \neq y^n$ and $y \in \mathcal{Y}^{out}$, $(\boldsymbol{x}, y, y^n)$ is an open-set noise.

Specifically, we have $y \sim P(\mathrm{y} = y|\mathbf{x} = \boldsymbol{x}; \mathrm{y} \in \mathcal{Y}^{all})$ while $y^n \sim P^n(\mathrm{y} = y^n|\mathbf{x} = \boldsymbol{x}; \mathrm{y} \in \mathcal{Y}^{all})$.

However, we can only identify label noise type with $(\boldsymbol{x}, y, y^n)$ — $y$ yet to be sampled with unknown clean conditional probability $P(\mathrm{y} = y|\mathbf{x} = \boldsymbol{x}; \mathrm{y} \in \mathcal{Y}^{all})$. To enable sample-wise analysis on the impact of different label noise, we thus introduce below $(O_{\boldsymbol{x}}, C_{\boldsymbol{x}})$ *label noise*:

**Definition 3.3** ($(O_{\boldsymbol{x}}, C_{\boldsymbol{x}})$ label noise). For sample $\boldsymbol{x}$ with clean conditional probability $P(\mathrm{y}|\mathbf{x} = \boldsymbol{x}; \mathrm{y} \in \mathcal{Y}^{all})$ and complete noise transition matrix $T$:

$$O_{\boldsymbol{x}} = \sum_{i=A+1}^{A+B} \sum_{j=1}^{A} T_{ij} P(\mathrm{y} = i|\mathbf{x} = \boldsymbol{x}; \mathrm{y} \in \mathcal{Y}^{all}) = \sum_{i=A+1}^{A+B} P(\mathrm{y} = i|\mathbf{x} = \boldsymbol{x}; \mathrm{y} \in \mathcal{Y}^{all}),$$

$$C_{\boldsymbol{x}} = \sum_{i=1}^{A} \sum_{j=1, j \neq i}^{A} T_{ij} P(\mathrm{y} = i|\mathbf{x} = \boldsymbol{x}; \mathrm{y} \in \mathcal{Y}^{all}) = \sum_{i=1}^{A} (1 - T_{ii}) P(\mathrm{y} = i|\mathbf{x} = \boldsymbol{x}; \mathrm{y} \in \mathcal{Y}^{all}). \tag{2}$$

Here, $O_{\boldsymbol{x}}$ represents the expected open-set noise ratio, and $C_{\boldsymbol{x}}$ represents the expected closed-set noise ratio. We then define sample $\boldsymbol{x}$ as an $(O_{\boldsymbol{x}}, C_{\boldsymbol{x}})$ label noise - as per Definition 3.2, sample $\boldsymbol{x}$ is expected to be an open-set noise with probability as $O_{\boldsymbol{x}}$ and expected to be an open-set noise with probability $O_{\boldsymbol{x}}$.

With Definition 3.3, we further formalize the concept of noise ratio for the whole distribution:

**Definition 3.4** (Accumulated noise ratio). We define the accumulated noise ratio, $N$, as the accumulated $(O_{\boldsymbol{x}}, C_{\boldsymbol{x}})$ label noise over all sample points $\boldsymbol{x} \in \mathcal{X}$:

$$N = \int_{\boldsymbol{x}} N_{\boldsymbol{x}} \cdot P(\mathbf{x} = \boldsymbol{x}; \mathrm{y} \in \mathcal{Y}^{all}) d\boldsymbol{x} = \int_{\boldsymbol{x}} (O_{\boldsymbol{x}} + C_{\boldsymbol{x}}) \cdot P(\mathbf{x} = \boldsymbol{x}; \mathrm{y} \in \mathcal{Y}^{all}) d\boldsymbol{x} \tag{3}$$

Here, $N_{\boldsymbol{x}}$ is referred to as the *point-wise noise ratio*.

### 3.2 ANALYZING CLASSIFICATION ERROR RATE INFLATION IN LNL

In this section, we analyze the impact of different types of label noise. We emphasize that, while the reformulated LNL setting encompasses outlier classes $\mathcal{Y}^{out}$, both during training and evaluation, they are unknown(agnostic). In other words, the default classification evaluation protocol focuses solely on the classification error rate over the inlier classes; the learned model $f$ is still tailored for the classification of inlier classes $\mathcal{Y}^{in}$.

**Error rate inflation** Specifically, we denote as $P^f(\mathrm{y}|\mathbf{x} = \boldsymbol{x}; \mathrm{y} \in \mathcal{Y}^{in})$ the learned *inlier conditional probability* with model $f$. In the evaluation phase, for specific sample $\boldsymbol{x}$ the prediction is

given by: $y^f = \arg\max_k P^f(y = k|\mathbf{x} = \boldsymbol{x}; y \in \mathcal{Y}^{in}) \in \mathcal{Y}^{in}$, and the corresponding expected classification error rate is defined as:

$$E_{\boldsymbol{x}} = \sum_{y \neq y^f} P(y = y, \mathbf{x} = \boldsymbol{x}; y \in \mathcal{Y}^{in}) = (1 - P(y = y^f)|\mathbf{x}; y \in \mathcal{Y}^{in})) \cdot P(\mathbf{x} = \boldsymbol{x}; y \in \mathcal{Y}^{in}). \tag{4}$$

We also have the Bayes error rate corresponds to the Bayes optimal model $f^*$:

$$E_{\boldsymbol{x}}^* = (1 - \max_k P(y = k|\mathbf{x} = \boldsymbol{x}; y \in \mathcal{Y}^{in})) \cdot P(\mathbf{x} = \boldsymbol{x}; y \in \mathcal{Y}^{in}). \tag{5}$$

To measure the negative impact of noisy labels, we care about the additional errors introduced, measured by the *error rate inflation*:

**Definition 3.5** (Error rate inflation)**.** With $E_{\boldsymbol{x}}^*$ as the Bayes error rate, we define the *error rate inflation* for sample $\boldsymbol{x}$ as: $\Delta E_{\boldsymbol{x}} = E_{\boldsymbol{x}} - E_{\boldsymbol{x}}^*$.

**Two pragmatic cases**    However, $P^f(y|\mathbf{x} = \boldsymbol{x}; y \in \mathcal{Y}^{in})$, as the prediction of the learned model $f$, is influenced by many factors (such as model capacity, dataset size, training hyperparameters like the number of epochs, etc.), making it non-trivial to determine its exact value for offline analysis[2]. Therefore, we consider two specific pragmatic cases that encompass most learning scenarios:

- ***Fitted case***: the model perfectly fits the noisy distribution: $P^f(y|\mathbf{x} = \boldsymbol{x}; y \in \mathcal{Y}^{in}) = P^n(y|\mathbf{x} = \boldsymbol{x}; y \in \mathcal{Y}^{in})$. This case may occur in scenarios such as fine-tuning a linear classifier with a frozen pre-trained model - as the pre-trained model already captures well-separated sample representations and the capacity of a linear classifier is limited.

- ***Overfitted case***: the model completely memorises the noisy labels: $P^f(y|\mathbf{x} = \boldsymbol{x}; y \in \mathcal{Y}^{in}) = P^{y^n}(y|\mathbf{x} = \boldsymbol{x}; y \in \mathcal{Y}^{in})$ - here $P^{y^n}$ denotes the one-hot encoded noisy label $y^n$. This case may arise in scenarios such as training a standard deep neural network from scratch with a single-label dataset - where the model normally has sufficient capacity to memorize all the labels.

### 3.3 Error rate inflation analysis *w.r.t* different label noise

In this section, we focus on analyzing the error rate inflation caused by different types of label noise. Let us recall the clean conditional distribution as $P(y|\mathbf{x}; y \in \mathcal{Y}^{all})$. For ease of analysis, we consider a simple scenario, wherein the entire clean conditional distribution over $\mathcal{X}$ remains unchanged, except for one sample point, say $\boldsymbol{x}$, which is affected by label noise:

$$P^n(y|\mathbf{x} \neq \boldsymbol{x}; y \in \mathcal{Y}^{all}) = P(y|\mathbf{x} \neq \boldsymbol{x}; y \in \mathcal{Y}^{all}), \ P^n(y|\mathbf{x} = \boldsymbol{x}; y \in \mathcal{Y}^{all}) \neq P(y|\mathbf{x} = \boldsymbol{x}; y \in \mathcal{Y}^{all}).$$

Under this condition, we simplify the analysis of the impact of label noise on the entire distribution to analyzing the error rate inflation of a single sample $\boldsymbol{x}$. Let us denote $P(y|\mathbf{x} = \boldsymbol{x}; y \in \mathcal{Y}^{all}) = [p_1, ..., p_A, ..., p_{A+B}]$, and denote its noise transition matrix as $T = \{T_{ij}\}_{i,j=1}^{A+B}$.

*Remark* 3.6 (Derivation of $\Delta E_{\boldsymbol{x}}$)**.** The inflation of the error rate, $\Delta E_{\boldsymbol{x}}$, is dependent on the complete noise transition matrix $T$, and the clean conditional probability $[p_1, ..., p_A, ..., p_{A+B}]$. Specifically, for above two cases, we have the corresponding error rate inflation for sample $\boldsymbol{x}$ as follows:

- ***Fitted case***:
$$\Delta E_{\boldsymbol{x}} = \max[p_1, ..., p_A] - p_{\arg\max[\sum_{i=1}^{A+B} p_i T_{i1}, ..., \sum_{i=1}^{A+B} p_i T_{iA}]}$$

- ***Overfitted case***:
$$\Delta E_{\boldsymbol{x}} = \max[p_1, ..., p_A] - \sum_{i=1}^{A} \left(p_i \cdot \sum_{j=1}^{A+B} p_j T_{ji}\right)$$

For a detailed derivation, please refer to appendix C.

---

[2]Please refer to (Mohri et al., 2018) for more discussions about related topics such as model generalization.

**Comparative analysis with proxy samples $x_1$ and $x_2$**   For the subsequent comparative analysis, we consider two proxy sample points, $x_1$ and $x_2$, corresponding to the different scenarios being compared. Following the notation used for sample $x$, we add subscripts to denote samples $x_1$ and $x_2$. For example, for sample $x_1$, we have $P(\mathbf{y}|\mathbf{x} = x_1; \mathbf{y} \in \mathcal{Y}^{all}) = [p_1^1, ..., p_A^1, ..., p_{A+B}^1]$, and the complete noise transition matrix as $T^1 = \{T_{ij}^1\}_{i,j=1}^{A+B}$. To ensure a strict fair comparison, we analyze the impact of various noise and noise modes while maintaining a consistent overall noise ratio. Firstly, we assume the following:

$$O_{x_1} + C_{x_1} = O_{x_2} + C_{x_2}. \tag{6}$$

Intuitively, we compare the error rate inflation ($\Delta E_{x_1}$ vs. $\Delta E_{x_2}$) under different label noise conditions given the *same point-wise noise ratio*. Additionally, we assume that $x_1$ and $x_2$ have the same prior sampling probability: $P(\mathbf{x} = x_1; \mathbf{y} \in \mathcal{Y}^{all}) = P(\mathbf{x} = x_2; \mathbf{y} \in \mathcal{Y}^{all})$, so that samples $x_1$ and $x_2$ are *probabilistically interchangeable* during the training set sampling process. These two conditions together also ensure that the accumulated noise ratio $N$ remains unchanged.

### 3.3.1 How does open-set noise compare to closed-set noise?

We begin by elucidating the differences between open-set noise and closed-set noise — in particular, we are interested in understanding the effects of having "more open-set noise" versus "more closed-set noise", given that $O_x + C_x$ remains unchanged. Without loss of generality, we consider:

$$O_{x_1} > O_{x_2} \;,\; C_{x_1} < C_{x_2}. \tag{7}$$

Intuitively, we regard sample $x_1$ as being more prone to open-set noise compared to sample $x_2$, thus corresponding to the 'more open-set noise' scenario. However, without additional regularization, there are infinitely many solutions that satisfy eq. (6) and eq. (7). Given the specific $P(\mathbf{y}|\mathbf{x} = x_1; \mathbf{y} \in \mathcal{Y}^{all})$ and $P(\mathbf{y}|\mathbf{x} = x_2; \mathbf{y} \in \mathcal{Y}^{all})$, the corresponding noise transition matrices $T^1$ and $T^2$ (see the example below) may not be unique. Therefore, the analysis of $\Delta E_{x_1}$ versus $\Delta E_{x_2}$ is not feasible—according to Remark 3.6, the values of $\Delta E_{x_1}$ and $\Delta E_{x_2}$ cannot be determined.

> **Toy Example of Agnostic $T$**   Assume a ternary classification with two known inlier classes ("0" and "1") and one unknown outlier class "2". Consider a sample $x_1$ with clean conditional probability $[0.1, 0.2, 0.7]$. Now, assume two different noise transition matrices for $T^1$ as follows:
>
> $$[0.55, 0.45, 0.0] = [0.1, 0.2, 0.7] \left[ \begin{array}{cc|c} 0.5 & 0.5 & 0 \\ 0.75 & 0.25 & 0 \\ \hline 0.5 & 0.5 & 0 \end{array} \right]$$
>
> $$[0.45, 0.55, 0.0] = [0.1, 0.2, 0.7] \left[ \begin{array}{cc|c} 0 & 1 & 0 \\ 0.5 & 0.5 & 0 \\ \hline 0.5 & 0.5 & 0 \end{array} \right]$$
>
> In both conditions, we have $O_{x_1} = 0.7$ and $C_{x_1} = 0.2$, but we arrive at different noisy conditional probabilities, similarly for sample $x_2$.

We thus consider a class-concentrated assumption—in most classification datasets, that the majority of samples belong exclusively to a specific class with high probability. In this condition, we have proved:

**Theorem 3.7** (Open-set noise *vs* closed-set noise). *Consider samples $x_1$ and $x_2$ satisfying eq. (6) and eq. (7) — compared to $x_2$, $x_1$ is considered as more prone to open-set noise. Let us denote $a = \arg\max_i P(\mathbf{y} = i|\mathbf{x} = x_1; \mathbf{y} \in \mathcal{Y}^{all})$ and $b = \arg\max_i P(\mathbf{y} = i|\mathbf{x} = x_2; \mathbf{y} \in \mathcal{Y}^{all})$, and assume (with high probability): $p_a^1 \to 1, \{p_i^1 \to 0\}_{i \neq a}$ and $p_b^2 \to 1, \{p_b^2 \to 0\}_{i \neq b}$. Then, we have:*

$$\Delta E_{x_1} < \Delta E_{x_2}$$

*in both **Fitted case** and **Overfitted case**.*

Please refer to appendix D.1 for a detailed proof. *In summarize, we validate that in most conditions, open-set noise is less harmful than closed-set noise in both **fitted case** and **overfitted case**, regardless of the specific noise mode.*

### 3.3.2 How does different open-set noise compare to each other?

We further study how different types of open-set noise affect the model. Specifically, we focus on the impacts of different open-set noise modes ($T_{out}$) given the same open-set noise ratio:

$$O_{\boldsymbol{x}_1} = O_{\boldsymbol{x}_2}. \tag{8}$$

To focus on open-set noise only and exclude the effect of closed-set noise, we assume:

$$C_{\boldsymbol{x}_1} = C_{\boldsymbol{x}_2} = 0 \tag{9}$$

Especially, in this section, we assume sample $\boldsymbol{x}_1$ and sample $\boldsymbol{x}_2$ holds the same clean conditional probability: $[p_1^1, ..., p_A^1, ..., p_{A+B}^1] = [p_1^2, ..., p_A^2, ..., p_{A+B}^2]$, allowing us to focus only on the impact of different open-set noise modes ($T_{out}$). We abbreviate the superscripts for simplicity subsequently. According to Definition 3.3, it is straightforward that $O_{\boldsymbol{x}_1} = O_{\boldsymbol{x}_2}$ always holds since $\sum_{i=A+1}^{A+B} p_i^1 = \sum_{i=A+1}^{A+B} p_i^2$, and $C_{\boldsymbol{x}_1} = C_{\boldsymbol{x}_2} = 0$ when $T_{in}^1 = T_{in}^2 = \mathbf{I}$[3].

Then, we have the flexibility to explore various forms of $T_{out}$ — corresponding to different open-set noise modes. Specifically, we consider two distinct open-set noise modes: 'easy' open-set noise when the transition from outlier classes to inlier classes involves completely random flipping, and 'hard' open-set noise when there exists an exclusive transition between the outlier class and specific inlier class. We denote as $T^{easy}$ for 'easy' open-set noise and $T^{hard}$ for 'hard' open-set noise, with intuitive explanations below:

$$T^{easy} = \begin{bmatrix} \frac{1}{A} & \cdots & \frac{1}{A} \\ \cdots & \cdots & \cdots \\ \frac{1}{A} & \cdots & \frac{1}{A} \end{bmatrix}_{B \times A} \tag{10}$$

and

$$T^{hard} = \begin{bmatrix} 0 & \cdots & 1 \\ \cdots & \cdots & \cdots \\ 1 & \cdots & 0 \end{bmatrix}_{B \times A} \tag{11}$$

Especially, for $T^{easy}$, we have $T_{ij} = \frac{1}{A}$ everywhere; for $T^{hard}$, we denote as $H_i : \{\arg_j(T_{ji}^{hard} = 1)\}_{i=1}^A$ the set of corresponding outlier classes $j \in \mathcal{Y}^{out}$ confused to inlier class $i \in \mathcal{Y}^{in}$. We would like to reiterate that although this resembles the widely studied symmetric and asymmetric noise, here we do not further assume that different sample points follow the same noise transition matrix.

Without loss of generality, we consider $\boldsymbol{x}_1$ with 'easy' open-set noise $T^{easy}$ and $\boldsymbol{x}_2$ with 'hard' open-set noise $T^{hard}$. Please note, that we no longer require class concentration assumption here as the noise transition matrix is considered known. In this condition, we have proved:

**Theorem 3.8** ('Hard' open-set noise *vs* 'easy' open-set noise)**.** *Consider samples $\boldsymbol{x}_1$, $\boldsymbol{x}_2$ satisfying eq. (8) and eq. (9). We set the corresponding noise transition matrix as $T_{out}^1 = T^{easy}, T_{out}^2 = T^{hard}, T_{in}^1 = T_{in}^2 = \mathbf{I}$ and denote $P(\mathrm{y}|\mathbf{x} = \boldsymbol{x}_1; \mathrm{y} \in \mathcal{Y}^{all}) = P(\mathrm{y}|\mathbf{x} = \boldsymbol{x}_2; \mathrm{y} \in \mathcal{Y}^{all}) = [p_1, ..., p_A, ..., p_{A+B}]$. Then, we have:*

- *Fitted case:*

$$\Delta E_{\boldsymbol{x}_1} \le \Delta E_{\boldsymbol{x}_2}.$$

- *Overfitted case:*

$$\Delta E_{\boldsymbol{x}_1} - \Delta E_{\boldsymbol{x}_2} = \sum_{i=1}^A a_i b_i.$$

*where $a_i = p_i$ and $b_i = \sum_{j \in H_i} p_j - \frac{1}{A} \sum_{i=A+1}^{A+B} p_i$.*

Please refer to appendix D.2 for a detailed proof. Specifically, we further discuss about **overfitted case** here. Since $\sum_{i=1}^A b_i = 0, \sum_{i=1}^A a_i = 1$, we can easily infer $\max(\Delta E_{\boldsymbol{x}_1} - \Delta E_{\boldsymbol{x}_2}) \ge 0, \min(\Delta E_{\boldsymbol{x}_1} - \Delta E_{\boldsymbol{x}_2}) \le 0$. With *rearrangement inequality* (theorem D.3), we note when the ranking of $\{p_i\}_{i=1}^A$ is completely in agreement with the ranking $\{\sum_{j \in H_i} p_j\}_{i=1}^A$ (constant term $-\frac{1}{A} \sum_{i=A+1}^{A+B} p_i$ omitted here), we reach its maximum value with $\Delta E_{\boldsymbol{x}_1} - \Delta E_{\boldsymbol{x}_2} \ge 0$. Intuitively speaking, this implies a

---

[3]Please refer to appendix D.3 for the analysis with additional concurrent closed-set noise, i.e., $T_{in}^1 = T_{in}^2 \ne \mathbf{I}$.

scenario that the 'hard' open-set noise tends to confuse a sample into the inlier class it primarily belongs to (with higher semantic similarity), as indicated by its higher probability (the higher the $p_i$ the higher the $\sum_{j \in H_i} p_j$). For example, an outlier 'tiger' image is wrongly included as a 'cat' rather than a 'dog' in a 'cat *vs* dog' binary classification dataset. As this is more consistent with the common intuition for semantic hardness, we default to such noise mode for 'hard' open-set noise — assuming the ranking of $\{p_i\}_{i=1}^A$ is of high agreement with the ranking of $\{\sum_{j \in H_i} p_j\}_{i=1}^A$.

*To summarize, we notice the 'hard' open-set noise and the 'easy' open-set noise exhibit contrasting trends in two different cases. In the **fitted case**, 'easy' open-set noise appears to be less harmful, while in the **overfitted case**, the impact of 'hard' open-set noise is comparatively smaller.*

### 3.4 RETHINKING OPEN-SET NOISE DETECTION

In addition to examining the impact of various types of open-set label noise on model generalization, we also assess the performance of current learning with noisy labels (LNL) methods when confronted with different types of open-set noise. Most current LNL methods primarily address closed-set noise, while the few sample selection approaches that target open-set noise generally focus on 'easy' open-set noise only. In this section, we evaluate the effectiveness of current methods in handling different forms of open-set noise, including the newly introduced 'hard' open-set noise.

We specifically focus on an *entropy-based open-set noise detection* mechanism, which has been widely applied in prior out-of-distribution (OOD) detection works (Chan et al., 2021; Xing et al., 2024). Within the sample selection framework, several methods (Albert et al., 2022; Sachdeva et al., 2021) have sought to extend closed-set noise detection techniques to identify open-set noise based on similar principles. These methods are generally founded on the empirical observation that samples with less confident and more averaged predictions often correspond to open-set instances, as indicated by high entropy in the model's predictions.

Similar to general sample selection methods, entropy-based open-set noise detection also occurs in the early stages after the model has undergone a few epochs training, commonly referred to as model warm-up training. At this point, the model is expected to have learned meaningful information while avoiding overfitting. In this context, we assume that the current model used for open-set noise detection is consistent with the **fitted case** described earlier.

Specifically, for a given sample $x$, we consider three variants for comparison: the original sample without noise transition, treated as a clean sample ; the 'hard' open-set noise sample, following the $T^{hard}$ open-set noise mode; and the 'easy' open-set noise sample, following the $T^{easy}$ open-set noise mode. We denote the prediction entropy values corresponding to these three variants as $\mathcal{H}_{easy}$, $\mathcal{H}_{hard}$, and $\mathcal{H}_{clean}$, respectively, and we have[4]:

$$
\begin{aligned}
\mathcal{H}_{clean} &= \text{ENT}([\frac{p_1}{\sum_{i=1}^A p_i}, ..., \frac{p_A}{\sum_{i=1}^A p_i}]) \\
&= \text{ENT}([p_1 + \frac{p_1}{\sum_{i=1}^A p_i} \sum_{i=A+1}^{A+B} p_i, ..., p_A + \frac{p_A}{\sum_{i=1}^A p_i} \sum_{i=A+1}^{A+B} p_i]), \\
\mathcal{H}_{easy} &= \text{ENT}([p_1 + \frac{1}{A} \sum_{i=A+1}^{A+B} p_i, ..., p_A + \frac{1}{A} \sum_{i=A+1}^{A+B} p_i]), \\
\mathcal{H}_{hard} &= \text{ENT}([p_1 + \sum_{j \in H_1} p_j, ..., p_A + \sum_{j \in H_A} p_j]).
\end{aligned}
\tag{12}
$$

Obviously, we have $\mathcal{H}_{easy} \geq \mathcal{H}_{clean}$[5]. However, comparing $\mathcal{H}_{hard}$ and $\mathcal{H}_{clean}$ is non-trivial without specific values for each entry. *Thus, we propose that open-set noise detection based on prediction entropy may only be effective for 'easy' open-set noise. This also suggests that the current success of prior sample selection methods involving open-set noise may be constrained.*

---

[4]Derivation omitted as most steps are similar to the proof in appendix D.2, specifically eq. (34) and eq. (35).

[5]Empirically, the relative scarcity of open-set noise can also lead to low-confidence/high-entropy predictions, a phenomenon beyond the scope of this work. We leave this for future exploration by interested readers.

## 4 EXPERIMENTS

In this section, we aim to validate our theoretical findings. In section 4.1, we verify the theoretical comparisons of different types of label noise. In section 4.2, we examine the entropy dynamics under different open-set label noise modes. Furthermore, in appendix E, we revisit the performance of current LNL methods in dealing with different open-set noise, including real-world WebVision noisy dataset. Additionally, we experiment with robust loss functions under varying open-set noise settings in appendix F. Finally, in appendix G, we propose several potential solutions for identifying and learning with open-set noise and conduct preliminary experiments.

To conduct more controlled, fair, and accurate experiments, we introduce two synthetic open-set noisy datasets—CIFAR100-O and ImageNet-O—constructed from the CIFAR100 and ImageNet datasets, respectively. Furthermore, we introduce a novel open-set test set for the widely used WebVision benchmark. For more details on datasets and implementations, please refer to appendix A.

### 4.1 EMPIRICAL VALIDATION OF THEOREM 3.7 AND THEOREM 3.8

In this section, we conduct experiments to validate the theorem 3.7 and theorem 3.8. Since most deep models have sufficient capacity, we consider direct supervised learning from scratch on the noisy dataset, treating the final model as the ***overfitted case*** - as evidenced by nearly $100\%$ classification accuracy on the training set. Conversely, obtaining a model that perfectly fits the data distribution is often challenging; here, we consider training a single-layer linear classifier upon a frozen pretrained encoder. Due to the limited capacity of the linear layer, we expect it to approximate the ***fitted case***.

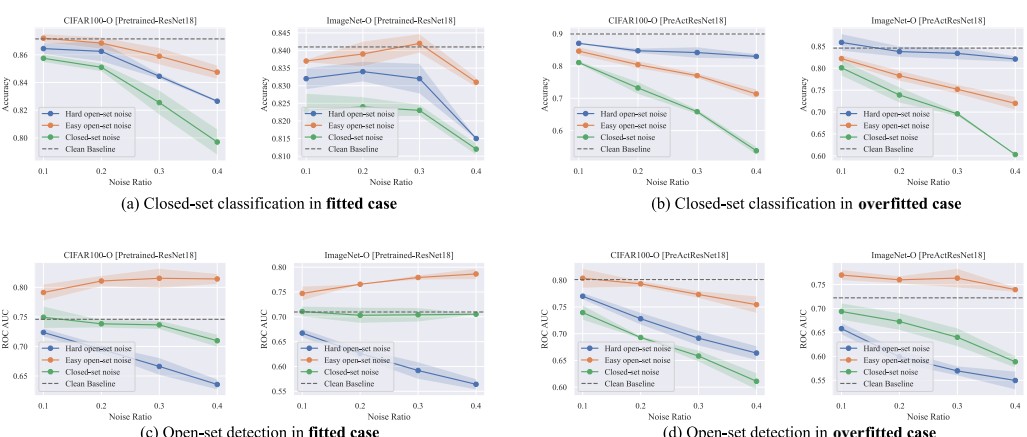

Figure 2: Direct supervised training with different noise modes and noise ratios. 'PreActResNet18' corresponds to ***overfitted case*** while 'Pretrained-ResNet18' corresponds to ***fitted case***.

We present the classification accuracy, i.e., 1 - classification error rate, on the CIFAR100-O and ImageNet-O datasets across different noise ratios, as shown in fig. 2(a/b). Regardless of the dataset or noise ratio, we observe that: (1) In both cases, the impact of open-set noise on classification accuracy is much smaller compared to closed-set noise. (2) 'Hard' open-set noise and 'easy' open-set noise exhibit opposite trends under the two different cases. These results align with our theoretical analysis.

Since open-set noise has a relatively small impact on classification accuracy, evaluating accuracy alone may not fully capture the model's performance in handling open-set noise. Therefore, we also report the model's out-of-distribution (OOD) detection performance (Hendrycks and Gimpel, 2016), as shown in fig. 2(c/d). For more details on the OOD detection task[6], please refer to appendix A.3. We observe that in both cases, the presence of open-set noise degrades OOD detection performance, whereas, conversely, the presence of closed-set noise could even improve OOD detection performance. For example, we notice that in the fitted case, the existence of open-set noise leads to steady

---

[6]Please note the distinction between the OOD detection task here and the open-set noise detection mechanism mentioned in the methods section.

improvement in OOD detection performance for both CIFAR100-O and ImageNet-O datasets, across different noise ratios. Given this contrasting trend, we propose that beyond the default closed-set classification, alternative evaluation frameworks, such as OOD detection, may provide a more comprehensive assessment of LNL methods.

## 4.2 Inspecting entropy-based open-set noise detection mechanism

In section 3.4, we analyze the open-set noise detection mechanism based on the entropy values of model predictions and find that it may be effective only for 'easy' open-set noise. Here, we empirically validate this across different open-set noise ratios. Specifically, we follow the warm-up training strategy, training on the entire dataset for a certain number of epochs. We then report the model's predicted entropy values for each sample at different warm-up epochs (5th, 10th, 20th) in fig. 3. Our results confirm that entropy dynamics serve as a more effective indicator for 'easy' open-set noise compared to 'hard' open-set noise ((a) vs (b), (c) vs (d) in fig. 3). We also test with mixed noise including both open-set noise and closed-set noise. For further discussion, please refer to appendix B.

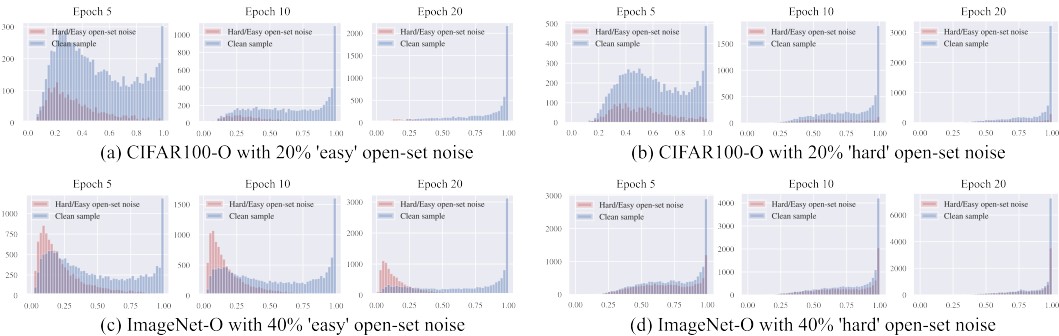

Figure 3: Entropy dynamics *w.r.t* different datasets, noise modes, and noise ratios.

## 5 Conclusions

This paper investigates the impact of open-set label noise on model performance. Although the "open world" setting, involving open-set samples, has been widely discussed in other weakly supervised learning contexts, its application in learning with noisy labels (LNL) remains underexplored. In response, we revisit the LNL problem, specifically examining the effects of open-set noise in comparison to closed-set noise, as well as the differences among various types of open-set noise in terms of classification performance. We find that open-set noise has a smaller impact on model classification performance compared to common closed-set noise, and different modes of open-set noise exhibit notable differences. Recognizing the limitations of existing evaluation frameworks in handling open-set noise, we explore the out-of-distribution (OOD) detection task to address shortcomings in model assessment and conduct preliminary experiments.

Additionally, we examine a common mechanism for detecting open-set noise based on prediction entropy, finding that it may only be effective for 'easy' open-set noise. Overall, our theoretical and empirical findings highlight the need for further investigation into open-set noise and its intricate effects on model performance.

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

# A    EXPERIMENT DETAILS

## A.1    DATASET DETAILS

Previous works involving open-set noise also attempted to build synthetic noisy datasets, typically treating different datasets as open-set noise for each other to construct synthetic noisy dataset (Sachdeva et al., 2021; Wu et al., 2021). In this scenario, potential domain gaps could affect a focused analysis of open-set noise. In this work, we propose selecting inlier/outlier classes from the same dataset to avoid this issue. Besides, in previous works, the consideration of open-set noise modes often focused on random flipping from outlier classes to all possible inlier classes, which is indeed the 'easy' open-set noise adopted in this paper. However, our theoretical analysis and experimental findings demonstrate that 'easy' open-set noise and 'hard' open-set noise exhibit distinct characteristics. Therefore, relying solely on experiments with 'easy' open-set noise is insufficient, emphasizing the necessity to explore and understand the complexities associated with different types of open-set noise.

**CIFAR100-O**    For the original CIFAR100 dataset, in addition to the commonly-used 100 fine classes, there exist 20 coarse classes each consisting of 5 fine classes. To build CIFAR100-O, we select one fine class from each coarse class as an inlier class (20 classes in total) while considering the remaining classes as outlier classes (80 classes in total). Then, we consider 'hard' and 'easy' open-set noise as below:

- 'Hard': Randomly selected samples from the outlier classs belonging to the same coarse class are introduced as open-set noise of the target class.
- 'Easy': Regardless of the target category, samples from the remaining categories are randomly introduced as open-set noise.

**ImageNet-O**    For a more challenging benchmark, we consider ImageNet-1K datasets - consisting of 1,000 classes. Specifically, we randomly select 20 classes and artificially identify another 20 classes similar to each of them as outlier classes (paired by ranking):

*inliers= ['tench', 'great white shark', 'cock', 'indigo bunting', 'European fire salamander', 'African crocodile', 'barn spider', 'macaw', 'rock crab', 'golden retriever', 'wood rabbit', 'gorilla', 'abaya', 'beer bottle', 'bookcase', 'cassette player', 'coffee mug', 'shopping basket', 'trifle', 'meat loaf']*

*outliers= ['goldfish', 'tiger shark', 'hen', 'robin', 'common newt', 'American alligator', 'garden spider', 'sulphur-crested cockatoo', 'king crab', 'Labrador retriever', 'Angora', 'chimpanzee', 'academic gown', 'beer glass', 'bookshop', 'CD player', 'coffeepot', 'shopping cart', 'ice cream', 'pizza']*

Then, we consider 'hard' and 'easy' open-set noise as below:

- 'Hard': Randomly selected samples from similar outlier classes are introduced as open-set noise for the target category.
- 'Easy': Regardless of the target category, samples from the outlier classes are randomly introduced as open-set noise.

For OOD detection, we directly use the corresponding test sets of outlier classes from the original datasets.

**WebVision**    WebVision (Li et al., 2017) is a large-scale dataset comprising 1,000 image classes obtained through web crawling, which includes a substantial amount of open-set noise. Consistent with previous studies (Jiang et al., 2018; Li et al., 2020; Ortego et al., 2021), we evaluate our methods using the first 50 classes from the Google Subset of WebVision.

To assess the performance of out-of-distribution (OOD) detection on the WebVision dataset, we create a separate test set of open-set images, following the same collection process as the original dataset. Specifically, we use the Google search engine with class names as keywords and identify retrieved OOD samples that are not included in the training set for this test set.

**Closed-set noise**   We also evaluate closed-set noise in some experiments, and by default, we consider the commonly used symmetric closed-set noise for simplicity. It is important to note that in our theoretical analysis, we do not impose any specific assumptions about the form of closed-set noise; our results apply to both symmetric and asymmetric closed-set noise.

## A.2   IMPLEMENTATION DETAILS

In this section, we provide detailed implementation specifications for the experiments in section 4.1. We also briefly introduce the applied out-of-distribution (OOD) detection protocol.

**Fitted case**   For the *fitted case*, we train a randomly initialized classifier consisting of a single linear layer built on top of the ResNet18 encoder with pretrained weights. In the CIFAR100-O dataset experiments, a weak augmentation strategy, including image padding and random cropping, is applied during training, with a batch size of 512. The weight decay is set to 0.0005, and the model is trained for 100 epochs with a learning rate of 0.02, following a cosine annealing schedule.

For the ImageNet-O dataset, no augmentation is applied during training. The batch size remains 512, with a weight decay of 0.01. The model is similarly trained for 100 epochs with a learning rate of 0.02, following the same cosine annealing schedule.

**Overfitted case**   For the *overfitted case*, we train a PreResNet18 model from scratch. For both datasets, a weak augmentation strategy, including image padding and random cropping, is applied during training with a batch size of 128. The weight decay is set to 0.0005, and the model is trained for 200 epochs with a learning rate of 0.02, following a cosine annealing schedule.

## A.3   OOD DETECTION EVALUATION PROTOCOL

We employ the maximum softmax probability as proposed by Hendrycks and Gimpel (2016) for out-of-distribution (OOD) detection. Specifically, let the trained model $f$ output a softmax vector $\boldsymbol{p}_i$ for each sample $\boldsymbol{x}_i$. A threshold value $t$, ranging between 0 and 1, is selected. For evaluation, we assign binary labels to indicate whether a sample belongs to a known class (closed-set) or an unknown class (open-set), transforming the OOD detection task into a binary classification problem. Samples with a maximum softmax value $p_i^{\max}$ below the threshold are considered potential open-set examples, as a low maximum value suggests that the model has low confidence in assigning the sample to any specific class.

## B   ENTROPY DYNAMICS FOR MIXED LABEL NOISE

In addition to the open-set noise only scenario, we also examine the entropy dynamics with mixed label noise in fig. 4. The notation '0.2all_0.5easy' denotes a scenario where the overall noise ratio is 0.2, with half of the noise being classified as 'easy' open-set noise. In the case of mixed label noise, the presence of closed-set noise significantly complicates the detection of open-set noise. For instance, in fig. 4(d), the entropy values of open-set noise even surpass those of clean samples. Although not formally analyzed, this observation suggests that entropy dynamics, derived from model predictions, may be fragile, warranting a more cautious approach to handling open-set noise.

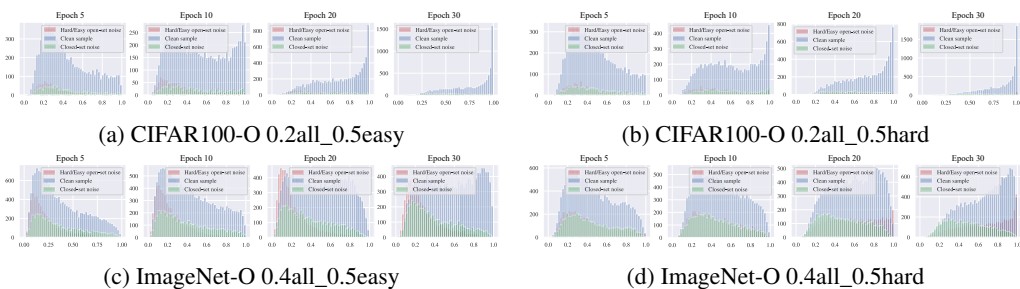

(a) CIFAR100-O 0.2all_0.5easy                    (b) CIFAR100-O 0.2all_0.5hard

(c) ImageNet-O 0.4all_0.5easy                    (d) ImageNet-O 0.4all_0.5hard

Figure 4: Entropy dynamics *w.r.t* mixed label noise.

## C  ERROR RATE INFLATION IN TWO DIFFERENT CASES

In this section, we present the computation details of error rate inflation in two interested cases - ***fitted case*** and ***overfitted case***. Specifically, we have:

- ***Fitted case***:

$$E_{\boldsymbol{x}} = (1 - P(\mathrm{y} = \arg\max_k P^n(\mathrm{y} = k|\mathbf{x} = \boldsymbol{x}; \mathrm{y} \in \mathcal{Y}^{in})|\mathbf{x} = \boldsymbol{x}; \mathrm{y} \in \mathcal{Y}^{in})) \cdot P(\mathbf{x} = \boldsymbol{x}; \mathrm{y} \in \mathcal{Y}^{in}). \tag{13}$$

- ***Overfitted case***:

$$\begin{aligned} E_{\boldsymbol{x}} &= (1 - P(\mathrm{y} = \arg\max_k P^{y^n}(\mathrm{y} = k|\mathbf{x} = \boldsymbol{x}; \mathrm{y} \in \mathcal{Y}^{in})|\mathbf{x} = \boldsymbol{x}; \mathrm{y} \in \mathcal{Y}^{in})) \cdot P(\mathbf{x} = \boldsymbol{x}; \mathrm{y} \in \mathcal{Y}^{in}) \\ &= \sum_{y^n \in \mathcal{Y}^{in}} (1 - P(\mathrm{y} = y^n|\mathbf{x} = \boldsymbol{x}; \mathrm{y} \in \mathcal{Y}^{in}))P^n(\mathrm{y} = y^n|\mathbf{x} = \boldsymbol{x}; \mathrm{y} \in \mathcal{Y}^{in}) \cdot P(\mathbf{x} = \boldsymbol{x}; \mathrm{y} \in \mathcal{Y}^{in}) \\ &= [1 - \sum_{y^n \in \mathcal{Y}^{in}} P(\mathrm{y} = y^n|\mathbf{x} = \boldsymbol{x}; \mathrm{y} \in \mathcal{Y}^{in})P^n(\mathrm{y} = y^n|\mathbf{x} = \boldsymbol{x}; \mathrm{y} \in \mathcal{Y}^{in})] \cdot P(\mathbf{x} = \boldsymbol{x}; \mathrm{y} \in \mathcal{Y}^{in}) \end{aligned} \tag{14}$$

While $E_{\boldsymbol{x}}^*$ denotes the Bayes optimal error rate:

$$E_{\boldsymbol{x}}^* = (1 - \max_k P(\mathrm{y} = k|\mathbf{x} = \boldsymbol{x}; \mathrm{y} \in \mathcal{Y}^{in})) \cdot P(\mathbf{x} = \boldsymbol{x}; \mathrm{y} \in \mathcal{Y}^{in}). \tag{15}$$

We thus have $\Delta E_{\boldsymbol{x}}$ in different cases as:

- ***Fitted case***:

$$\begin{aligned} \Delta E_{\boldsymbol{x}} &= \big(\max_k P(\mathrm{y} = k|\mathbf{x} = \boldsymbol{x}; \mathrm{y} \in \mathcal{Y}^{in}) - P(\mathrm{y} = \arg\max_k P^n(\mathrm{y} = k|\mathbf{x} = \boldsymbol{x}; \mathrm{y} \in \mathcal{Y}^{in})|\mathbf{x} = \boldsymbol{x}; \mathrm{y} \in \mathcal{Y}^{in})\big) \\ &\quad \cdot P(\mathbf{x} = \boldsymbol{x}; \mathrm{y} \in \mathcal{Y}^{in}). \end{aligned} \tag{16}$$

- ***Overfitted case***:

$$\begin{aligned} \Delta E_{\boldsymbol{x}} &= \big(\max_k P(\mathrm{y} = k|\mathbf{x} = \boldsymbol{x}; \mathrm{y} \in \mathcal{Y}^{in}) - \sum_{y^n \in \mathcal{Y}^{in}} P(\mathrm{y} = y^n|\mathbf{x} = \boldsymbol{x}; \mathrm{y} \in \mathcal{Y}^{in})P^n(\mathrm{y} = y^n|\mathbf{x} = \boldsymbol{x}; \mathrm{y} \in \mathcal{Y}^{in})\big) \\ &\quad \cdot P(\mathbf{x} = \boldsymbol{x}; \mathrm{y} \in \mathcal{Y}^{in}). \end{aligned} \tag{17}$$

**Details on the derivation of error rate inflation**  For better clarity, we here restate the notations in section 3.3. Let us denote $P(\mathrm{y}|\mathbf{x} = \boldsymbol{x}; \mathrm{y} \in \mathcal{Y}^{all}) = [p_1, ..., p_A, ..., p_{A+B}]$, and denote its noise transition matrix as $T = \{T_{ij}\}_{i,j=1}^{A+B}$. Here, $\{T_{ij} = 0\}$ for all $j > A$.

With eq. (1), we compute the corresponding noisy conditional probability as:

$$P^n(\mathrm{y}|\mathbf{x} = \boldsymbol{x}; \mathrm{y} \in \mathcal{Y}^{all}) = [\sum_{i=1}^{A+B} p_i T_{i1}, ..., \sum_{i=1}^{A+B} p_i T_{iA}, 0, ..., 0]. \tag{18}$$

We also have:

$$P(\mathrm{y} = k | \mathbf{x} = \boldsymbol{x}; \mathrm{y} \in \mathcal{Y}^{in}) = \frac{P(\mathrm{y} = k | \mathbf{x} = \boldsymbol{x}; \boldsymbol{y} \in \mathcal{Y}^{all})}{\sum_{i \in \mathcal{Y}^{in}} P(\mathrm{y} = i | \mathbf{x} = \boldsymbol{x}; \mathrm{y} \in \mathcal{Y}^{all})} = \frac{p_k}{\sum_{i=1}^{A} p_i},$$

$$P^n(\mathrm{y} = k | \mathbf{x} = \boldsymbol{x}; \mathrm{y} \in \mathcal{Y}^{in}) = \frac{P^n(\mathrm{y} = k | \mathbf{x} = \boldsymbol{x}; \boldsymbol{y} \in \mathcal{Y}^{all})}{\sum_{i \in \mathcal{Y}^{in}} P^n(\mathrm{y} = i | \mathbf{x} = \boldsymbol{x}; \mathrm{y} \in \mathcal{Y}^{all})} = \sum_{i=1}^{A+B} p_i T_{ik},$$

$$P(\mathbf{x} = \boldsymbol{x}; \mathrm{y} \in \mathcal{Y}^{in}) = \frac{\sum_{y \in \mathcal{Y}^{in}} P(\mathbf{x} = \boldsymbol{x}, \mathrm{y} = y; \mathrm{y} \in \mathcal{Y}^{all})}{\int \sum_{y \in \mathcal{Y}^{in}} P(\mathbf{x} = \boldsymbol{x}, \mathrm{y} = y; \mathrm{y} \in \mathcal{Y}^{all}) d\boldsymbol{x}} \tag{19}$$

$$\propto \sum_{y \in \mathcal{Y}^{in}} P(\mathbf{x} = \boldsymbol{x}, \mathrm{y} = y; \mathrm{y} \in \mathcal{Y}^{all})$$

$$\propto \sum_{y \in \mathcal{Y}^{in}} P(\mathrm{y} = y | \mathbf{x} = \boldsymbol{x}; \mathrm{y} \in \mathcal{Y}^{all}) P(\mathbf{x} = \boldsymbol{x}; \mathrm{y} \in \mathcal{Y}^{all})$$

$$\propto \sum_{i=1}^{A} p_i \cdot P(\mathbf{x} = \boldsymbol{x}; \mathrm{y} \in \mathcal{Y}^{all}).$$

Wrapping the above together, we have:

$$P(\mathrm{y} | \mathbf{x} = \boldsymbol{x}; \mathrm{y} \in \mathcal{Y}^{in}) = [\frac{p_1}{\sum_{i=1}^{A} p_i}, ..., \frac{p_A}{\sum_{i=1}^{A} p_i}],$$

$$P^n(\mathrm{y} | \mathbf{x} = \boldsymbol{x}; \mathrm{y} \in \mathcal{Y}^{in}) = [\sum_{i=1}^{A+B} p_i T_{i1}, ..., \sum_{i=1}^{A+B} p_i T_{iA}], \tag{20}$$

$$P(\mathbf{x} = \boldsymbol{x}; \mathrm{y} \in \mathcal{Y}^{in}) \propto \sum_{i=1}^{A} p_i \cdot P(\mathbf{x} = \boldsymbol{x}; \mathrm{y} \in \mathcal{Y}^{all}).$$

With eq. (16), eq. (17) and eq. (20), we can then compute and compare $\Delta E_{\boldsymbol{x}}$ in both ***fitted case*** and ***overfitted case***:

- **Fitted case**:

$$\boxed{\Delta E_{\boldsymbol{x}} = \big( \max[p_1, ..., p_A] - p_{\arg \max[\sum_{i=1}^{A+B} p_i T_{i1}, ..., \sum_{i=1}^{A+B} p_i T_{iA}]} \big) \cdot P(\mathbf{x} = x; \mathrm{y} \in \mathcal{Y}^{all})} \tag{21}$$

- **Overfitted case**:

$$\boxed{\Delta E_{\boldsymbol{x}} = \big( \max[p_1, ..., p_A] - \sum_{i=1}^{A} (p_i \cdot \sum_{j=1}^{A+B} p_j T_{ji}) \big) \cdot P(\mathbf{x} = x; \mathrm{y} \in \mathcal{Y}^{all})} \tag{22}$$

In the main section 3.2, we have omitted the sampling prior term $P(\mathbf{x} = \boldsymbol{x}; \mathrm{y} \in \mathcal{Y}^{all})$ (marked in gray) for simplicity — cause in our subsequent comparative analysis, we assume that the *sampling prior*: $P(\mathbf{x} = \boldsymbol{x}; \mathrm{y} \in \mathcal{Y}^{all})$ of the sample points is fixed to ensure a fair comparison. Please continue the section **Comparative analysis with proxy samples $\boldsymbol{x}_1$ and $\boldsymbol{x}_2$** for further explanation.

# D    FULL PROOF OF THEOREM 3.7 AND THEOREM 3.8

**Error rate inflation comparison *s.t.* same noise ratio**    To ensure a fair comparison, in this work, we focus on the impact of different label noise given the same noise ratio - modifying $O_{\boldsymbol{x}}$ and $C_{\boldsymbol{x}}$ while analyzing the trend of $\Delta E_{\boldsymbol{x}}$. Specifically, for two proxy sample points $\boldsymbol{x}_1$ and $\boldsymbol{x}_2$, we assume:

$$O_{\boldsymbol{x}_1} + C_{\boldsymbol{x}_1} = O_{\boldsymbol{x}_2} + C_{\boldsymbol{x}_2}. \tag{23}$$

which leads us to:

$$\sum_{i=A+1}^{A+B} p_i^1 + \sum_{i=1}^{A} \sum_{j=1,j\neq i}^{A} T_{ij}^1 p_i^1 = \sum_{i=A+1}^{A+B} p_i^2 + \sum_{i=1}^{A} \sum_{j=1,j\neq i}^{A} T_{ij}^2 p_i^2 \longrightarrow \sum_{i=1}^{A} T_{ii}^1 p_i^1 = \sum_{i=1}^{A} T_{ii}^2 p_i^2 \quad (24)$$

*Note that the superscript here refers to the sample point $\boldsymbol{x}_2$, not a square exponent.*

### D.1 Proof of theorem 3.7 — Open-set noise vs Closed-set noise

In this section, we try to compare open-set noise and closed-set noise. Without loss of generality, we consider:

$$O_{\boldsymbol{x}_1} > O_{\boldsymbol{x}_2}. \quad (25)$$

As clarified by the toy example in section 3.3.1, without extra regularizations, the noise transition matrix is not identifiable. *We thus consider a simple compromise situation - in most classification problems, the majority of samples (with a high probability) belong to a specific class exclusively with high probability.*

Let us denote:

$$a = \arg\max_i P(\mathrm{y} = i | \mathbf{x} = \boldsymbol{x}_1; \mathrm{y} \in \mathcal{Y}^{all})$$

and

$$b = \arg\max_i P(\mathrm{y} = i | \mathbf{x} = \boldsymbol{x}_2; \mathrm{y} \in \mathcal{Y}^{all}).$$

We assume :

$$p_a^1 \to 1, \{p_i^1 \to 0\}_{i\neq a}, p_b^2 \to 1, \{p_i^2 \to 0\}_{i\neq b},$$

and we have:

$$O_{\boldsymbol{x}_1} = \sum_{i=A+1}^{A+B} p_i^1, \ O_{\boldsymbol{x}_2} = \sum_{i=A+1}^{A+B} p_i^2.$$

With eq. (25), we easily infer that: $a \in \mathcal{Y}^{out}$ while $b \in \mathcal{Y}^{in}$. With eq. (24), we further have:

$$\sum_{i=1}^{A} T_{ii}^1 p_i^1 \approx \sum_{i=1}^{A} T_{ii}^1 \times 0 \approx 0,$$

$$\sum_{i=1}^{A} T_{ii}^2 p_i^2 \approx \sum_{i=1,i\neq b}^{A} T_{ii}^2 \times 0 + T_{bb}^2 \times 1 \approx T_{bb}^2.$$

Thus we have: $T_{bb}^2 \approx 0$, which enables us to analyze and compare $\Delta E_{\boldsymbol{x}_1}$ and $\Delta E_{\boldsymbol{x}_2}$:

**Fitted case** In this case, according to eq. (21), we have:

$$\begin{aligned}
\Delta E_{\boldsymbol{x}_1} &= \max[p_1^1,...,p_A^1] - p_{\arg\max[\sum_{i=1}^{A+B} p_i^1 T_{i1}^1,...,\sum_{i=1}^{A+B} p_i^1 T_{iA}^1]} \\
&\leq \max[p_1^1,...,p_A^1] - \min[p_1^1,...,p_A^1] \\
&\xrightarrow{p_a^1 \to 1, \{p_i^1 \to 0\}_{i\neq a}, a\in\mathcal{Y}^{out}} \\
&\approx 0,
\end{aligned} \quad (26)$$

$$\begin{aligned}
\Delta E_{\boldsymbol{x}_2} &= \max[p_1^2,...,p_A^2] - p_{\arg\max[\sum_{i=1}^{A+B} p_i^2 T_{i1}^2,...,\sum_{i=1}^{A+B} p_i^2 T_{iA}^2]} \\
&\xrightarrow{[\sum_{i=1}^{A+B} p_i^2 T_{i1}^2,...,\sum_{i=1}^{A+B} p_i^2 T_{iA}^2]\approx[T_{b1}^2, T_{b2}^2,..., \overbrace{0}^{T_{bb}^2},...,T_{bA}^2]} \\
&= p_b^2 - p_{n\neq b}^2 \\
&\xrightarrow{p_b^2 \to 1, \{p_i^2 \to 0\}_{i\neq b}, b\in\mathcal{Y}^{in}} \\
&\approx 1.
\end{aligned} \quad (27)$$

**Overfitted case** In this case, according to eq. (22), we similarly have:

$$\Delta E_{\boldsymbol{x}_1} = \max[p_1^1, ..., p_A^1] - \sum_{i=1}^{A}(p_i^1 \cdot \sum_{j=1}^{A+B} p_j^1 T_{ji}^1) \approx 0,$$

(28)

$$\Delta E_{\boldsymbol{x}_2} = \max[p_1^2, ..., p_A^2] - \sum_{i=1}^{A}(p_i^2 \cdot \sum_{j=1}^{A+B} p_j^2 T_{ji}^2) \approx 1.$$

(29)

We wrap up above for theorem D.1:

**Theorem D.1** (Open-set noise *vs* Closed-set noise). *Let us consider sample $\boldsymbol{x}_1$, $\boldsymbol{x}_2$ fulfilling eq. (23) and eq. (25) - compared to $\boldsymbol{x}_2$, $\boldsymbol{x}_1$ is considered as more prone to open-set noise. Let us denote $a = \arg\max_i P(\mathrm{y} = i|\mathbf{x} = \boldsymbol{x}_1; \mathrm{y} \in \mathcal{Y}^{all})$ and $b = \arg\max_i P(\mathrm{y} = i|\mathbf{x} = \boldsymbol{x}_2; \mathrm{y} \in \mathcal{Y}^{all})$, we assume (with a high probability): $p_a^1 \to 1, \{p_i^1 \to 0\}_{i \neq a}$ and $p_b^2 \to 1, \{p_b^2 \to 0\}_{i \neq b}$. Then, we have:*

$$\Delta E_{\boldsymbol{x}_1} < \Delta E_{\boldsymbol{x}_2}$$

*in both **fitted case** and **overfitted case**.*

### D.2 Derivation of theorem 3.7 — 'hard' open-set noise vs 'easy' open-set noise

In this part, we try to analyze and compare 'hard' open-set noise with 'easy' open-set noise. For better clarification, we repeat here the essential notations:

$$T_{out}^1 = T^{easy} = \begin{bmatrix} \frac{1}{A} & \cdots & \frac{1}{A} \\ \cdots & \cdots & \cdots \\ \frac{1}{A} & \cdots & \frac{1}{A} \end{bmatrix}_{B \times A}$$

(30)

and

$$T_{out}^2 = T^{hard} = \begin{bmatrix} 0 & \cdots & 1 \\ \cdots & \cdots & \cdots \\ 1 & \cdots & 0 \end{bmatrix}_{B \times A}$$

(31)

and

$$T_{in}^1 = T_{in}^2 = \mathbf{I}.$$

(32)

Especially, for $T^{easy}$, we have $T_{ij} = \frac{1}{A}$ everywhere; for $T^{hard}$, we denote as $H_i : \{\arg_j(T_{ji}^{hard} = 1)\}_{i=1}^{A}$ the set of corresponding outlier classes $j \in \mathcal{Y}^{out}$ confused to inlier class $i \in \mathcal{Y}^{in}$. We also have:

$$[p_1^1, ..., p_A^1, ..., p_{A+B}^1] = [p_1^2, ..., p_A^2, ..., p_{A+B}^2] = [p_1, ..., p_A, ..., p_{A+B}].$$

(33)

**Fitted case** In this case, according to eq. (21), for sample $\boldsymbol{x}_1$ with 'easy' open-set noise, we have:

$$\begin{aligned} \Delta E_{\boldsymbol{x}_1} &= \max[p_1^1, ..., p_A^1] - p_{\arg\max[\sum_{i=1}^{A+B} p_i^1 T_{i1}^1, ..., \sum_{i=1}^{A+B} p_i^1 T_{iA}^1]} \\ &= \max[p_1^1, ..., p_A^1] - p_{\arg\max[p_1^1 + \frac{1}{A}\sum_{i=A+1}^{A+B} p_i^1, ..., p_A^1 + \frac{1}{A}\sum_{i=A+1}^{A+B} p_i^1]} \\ &= 0, \end{aligned}$$

(34)

and, for sample $\boldsymbol{x}_2$ with 'hard' open-set noise, we have:

$$\begin{aligned} \Delta E_{\boldsymbol{x}_2} &= \max[p_1^2, ..., p_A^2] - p_{\arg\max[\sum_{i=1}^{A+B} p_i^2 T_{i1}^2, ..., \sum_{i=1}^{A+B} p_i^2 T_{iA}^2]} \\ &= \max[p_1^2, ..., p_A^2] - p_{\arg\max[p_1^2 + \sum_{b \in H_1} p_b^2, ..., p_A^2 + \sum_{b \in H_A} p_b^2]} \\ &\in [0, \max[p_1^2, ..., p_A^2] - \min[p_1^2, ..., p_A^2]]. \end{aligned}$$

(35)

**Overfitted case**   In this case, according to eq. (22), for sample $\boldsymbol{x}_1$ with 'easy' open-set noise, we have:

$$\Delta E_{\boldsymbol{x}_1} = \max[p_1^1, ..., p_A^1] - \sum_{i=1}^{A}(p_i^1 \cdot \sum_{j=1}^{A+B} p_j^1 T_{ji}^1)$$

$$= \max[p_1^1, ..., p_A^1] - \sum_{i=1}^{A}\left(p_i^1 \cdot (\sum_{j=1}^{A} p_j^1 T_{ji}^1 + \sum_{j=A+1}^{A+B} p_j^1 T_{ji}^1)\right) \tag{36}$$

$$\xrightarrow{T_{in}^1 = \mathbf{I},\ T_{out}^1 = T^{easy}}$$

$$= \max[p_1^1, ..., p_A^1] - \sum_{i=1}^{A} p_i^1(p_i^1 + \frac{1}{A}\sum_{i=A+1}^{A+B} p_i^1).$$

and, for sample $\boldsymbol{x}_2$ with 'hard' open-set noise, we have:

$$\Delta E_{\boldsymbol{x}_2} = \max[p_1^2, ..., p_A^2] - \sum_{i=1}^{A}(p_i^2 \cdot \sum_{j=1}^{A+B} p_j^2 T_{ji}^2)$$

$$= \max[p_1^2, ..., p_A^2] - \sum_{i=1}^{A}\left(p_i^2 \cdot (\sum_{j=1}^{A} p_j^2 T_{ji}^2 + \sum_{j=A+1}^{A+B} p_j^2 T_{ji}^2)\right) \tag{37}$$

$$\xrightarrow{T_{in}^2 = \mathbf{I},\ T_{out}^2 = T^{hard}}$$

$$= \max[p_1^2, ..., p_A^2] - \sum_{i=1}^{A} p_i^2(p_i^2 + \sum_{j \in H_i} p_j^2)$$

Omitting superscripts (eq. (33)), we further have:

$$\Delta E_{\boldsymbol{x}_1} - \Delta E_{\boldsymbol{x}_2} = \sum_{i=1}^{A} p_i(\sum_{j \in H_i} p_j - \frac{1}{A}\sum_{i=A+1}^{A+B} p_i). \tag{38}$$

Let $a_i = p_i, b_i = \sum_{j \in H_i} p_j - \frac{1}{A}\sum_{i=A+1}^{A+B} p_i$, we have:

$$\Delta E_{\boldsymbol{x}_1} - \Delta E_{\boldsymbol{x}_2} = \sum_{i=1}^{A} a_i b_i.$$

To summarize, we wrap up the above together:

**Theorem D.2** ('Hard' open-set noise *vs* 'easy' open-set noise). *Let us consider sample $\boldsymbol{x}_1$, $\boldsymbol{x}_2$ fulfilling eq. (8) and eq. (9). We set the corresponding noise transition matrix as $T_{out}^1 = T^{easy}, T_{out}^2 = T^{hard}, T_{in}^1 = T_{in}^2 = \mathbf{I}$ and denote $P(\mathrm{y}|\mathbf{x} = \boldsymbol{x}_1; \mathrm{y} \in \mathcal{Y}^{all}) = P(\mathrm{y}|\mathbf{x} = \boldsymbol{x}_2; \mathrm{y} \in \mathcal{Y}^{all}) = [p_1, ..., p_A, ..., p_{A+B}]$. Then, we have:*

• *Fitted case:*
$$\Delta E_{\boldsymbol{x}_1} \leq \Delta E_{\boldsymbol{x}_2}.$$

• *Overfitted case:*

$$\Delta E_{\boldsymbol{x}_1} - \Delta E_{\boldsymbol{x}_2} = \sum_{i=1}^{A} a_i b_i.$$

   *Here, $a_i = p_i, b_i = \sum_{j \in H_i} p_j - \frac{1}{A}\sum_{i=A+1}^{A+B} p_i$.*

**Theorem D.3** (Rearrangement Inequality). *For the sequences $a_1, a_2, \ldots, a_n$ and $b_1, b_2, \ldots, b_n$, where $a_1 \leq a_2 \leq \ldots \leq a_n$ and $b_1 \leq b_2 \leq \ldots \leq b_n$, the rearrangement inequality is given by:*
$$a_1 \cdot b_1 + a_2 \cdot b_2 + \ldots + a_n \cdot b_n \geq a_1 \cdot b_{\sigma(1)} + a_2 \cdot b_{\sigma(2)} + \ldots + a_n \cdot b_{\sigma(n)} \geq a_1 \cdot b_n + a_2 \cdot b_{n-1} + \ldots + a_n \cdot b_1$$
*Here, $\sigma$ denotes a permutation of the indices $1, 2, \ldots, n$. The leftmost expression corresponds to the case where $\sigma(i) = i$ (identity permutation), and the rightmost expression corresponds to the case where $\sigma(i) = n + 1 - i$ (reverse permutation).*

### D.3 'HARD' OPEN-SET NOISE *vs.* 'EASY' OPEN-SET NOISE WHEN $T_{in}^1 = T_{in}^2 \neq \mathbf{I}$

In the previous section, we analyzed open-set noise by setting the closed-set noise to zero to simplify the analysis. In this section, we relax this assumption and no longer assume $T_{in}^1 = T_{in}^2 \neq \mathbf{I}$. Intuitively, we aim to investigate whether the presence of additional closed-set noise affects the conclusions drawn earlier.

**Fitted case** we first investigate the fitted case. Similarly, we have:

$$
\begin{aligned}
\Delta E_{x_1} &= \max[p_1^1, \ldots, p_A^1] - p_{\arg\max[\sum_{i=1}^{A+B} p_i^1 T_{i1}^1, \ldots, \sum_{i=1}^{A+B} p_i^1 T_{iA}^1]} \\
&= \max[p_1^1, \ldots, p_A^1] - p_{\arg\max[\sum_{i=1}^{A} p_i^1 T_{i1} + \frac{1}{A}\sum_{i=A+1}^{A+B} p_i^1, \ldots, \sum_{i=1}^{A} p_i^1 T_{iA} + \frac{1}{A}\sum_{i=A+1}^{A+B} p_i^1]} \quad (39) \\
&= \max[p_1^1, \ldots, p_A^1] - p_{\arg\max[\sum_{i=1}^{A} p_i^1 T_{i1}, \ldots, \sum_{i=1}^{A} p_i^1 T_{iA}]}
\end{aligned}
$$

$$
\begin{aligned}
\Delta E_{x_2} &= \max[p_1^2, \ldots, p_A^2] - p_{\arg\max[\sum_{i=1}^{A+B} p_i^2 T_{i1}^2, \ldots, \sum_{i=1}^{A+B} p_i^2 T_{iA}^2]} \\
&= \max[p_1^2, \ldots, p_A^2] - p_{\arg\max[\sum_{i=1}^{A} p_i^2 T_{i1} + \sum_{b\in H_1} p_b^2, \ldots, \sum_{i=1}^{A} p_i^2 T_{iA} + \sum_{b\in H_A} p_b^2]}.
\end{aligned} \quad (40)
$$

Unfortunately, without extra assumptions on $T_{in}$ or $[p_1^1, \ldots, p_A^1]$, to compare $\Delta E_{x_1}$ and $\Delta E_{x_2}$ is impossible. Here, we consider two conservative but realistic cases:

**i. Concentration assumption of** $[p_1^1, \ldots, p_A^1]$: in this case, we assume the probability $[p_1^1, \ldots, p_A^1]$ concentrate on one specific class, say, $t$. We thus have $p_t^1 \to 1, p_k^1 \to 0, \forall k \neq t$. In this case, we have:

$$
\begin{aligned}
\Delta E_{x_1} &= \max[p_1^1, \ldots, p_A^1] - p_{\arg\max[\sum_{i=1}^{A} p_i^1 T_{i1}, \ldots, \sum_{i=1}^{A} p_i^1 T_{iA}]} \\
&\approx p_t^1 - p_{\arg\max[p_t^1 T_{t1}, \ldots, p_t^1 T_{tt}, \ldots, p_t^1 T_{tA}]} \\
&\xrightarrow{\text{diagnomal-dominant noise transition matrix}} \\
&= 0.
\end{aligned} \quad (41)
$$

$$
\begin{aligned}
\Delta E_{x_2} &= \max[p_1^2, \ldots, p_A^2] - p_{\arg\max[\sum_{i=1}^{A} p_i^2 T_{i1} + \sum_{b\in H_1} p_b^2, \ldots, \sum_{i=1}^{A} p_i^2 T_{iA} + \sum_{b\in H_A} p_b^2]} \\
&\approx p_t^2 - p_{\arg\max[p_t^2 T_{t1} + \sum_{b\in H_1} p_b^2, \ldots, p_t^2 T_{tt} + \sum_{b\in H_t} p_b^2, \ldots, p_t^2 T_{tA} + \sum_{b\in H_A} p_b^2]} \\
&\geq 0.
\end{aligned} \quad (42)
$$

Note we normally implciitly assume a daignomal-dominant noise transition matrix, that is, $\forall i, j \neq i, T_{ii} > T_{ij}$.

**ii. Symmetric closed-set noise for** $T_{in}$: in this case, we assume a symmetric noise transition matrix $T$.

$$
\begin{aligned}
\Delta E_{x_1} &= \max[p_1^1, \ldots, p_A^1] - p_{\arg\max[\sum_{i=1}^{A} p_i^1 T_{i1}, \ldots, \sum_{i=1}^{A} p_i^1 T_{iA}]} \\
&= \max[p_1^1, \ldots, p_A^1] - p_{\arg\max[\sigma + p_1^1 T_\Delta, \ldots, \sigma + p_A^1 T_\Delta]} \\
&= 0.
\end{aligned} \quad (43)
$$

$$
\begin{aligned}
\Delta E_{x_2} &= \max[p_1^2, \ldots, p_A^2] - p_{\arg\max[\sum_{i=1}^{A+B} p_i^2 T_{i1}^2, \ldots, \sum_{i=1}^{A+B} p_i^2 T_{iA}^2]} \\
&= \max[p_1^2, \ldots, p_A^2] - p_{\arg\max[\sum_{i=1}^{A} p_i^2 T_{i1} + \sum_{b\in H_1} p_b^2, \ldots, \sum_{i=1}^{A} p_i^2 T_{iA} + \sum_{b\in H_A} p_b^2]} \\
&= p_t^2 - p_{\arg\max[\sigma + p_1^2 T_\Delta + \sum_{b\in H_1} p_b^2, \ldots, \sigma + p_A^2 T_\Delta + \sum_{b\in H_A} p_b^2]} \\
&\geq 0.
\end{aligned} \quad (44)
$$

In above two cases, we still have $\Delta E_{x_1} \leq \Delta E_{x_2}$. That is to say, under either of the two popular assumptions above, we arrive at the same conclusion: 'easy' open-set noise is less harmful than 'hard' open-set noise.

**Overfitted case** we then re-investigate the overfitted-case. Similalrly, we have:

$$\Delta E_{x_1} = \max[p_1^1, ..., p_A^1] - \sum_{i=1}^{A}(p_i^1 \cdot \sum_{j=1}^{A+B} p_j^1 T_{ji}^1)$$

$$= \max[p_1^1, ..., p_A^1] - \sum_{i=1}^{A}\left(p_i^1 \cdot (\sum_{j=1}^{A} p_j^1 T_{ji}^1 + \sum_{j=A+1}^{A+B} p_j^1 T_{ji}^1)\right) \quad (45)$$

$$\xrightarrow{T_{in}^1 \neq \mathbf{I}, \ T_{out}^1 = T^{easy}}$$

$$= \max[p_1^1, ..., p_A^1] - \sum_{i=1}^{A} p_i^1 (\sum_{j=1}^{A} p_j^1 T_{ji}^1 + \frac{1}{A}\sum_{i=A+1}^{A+B} p_i^1).$$

$$\Delta E_{x_2} = \max[p_1^2, ..., p_A^2] - \sum_{i=1}^{A}(p_i^2 \cdot \sum_{j=1}^{A+B} p_j^2 T_{ji}^2)$$

$$= \max[p_1^2, ..., p_A^2] - \sum_{i=1}^{A}\left(p_i^2 \cdot (\sum_{j=1}^{A} p_j^2 T_{ji}^2 + \sum_{j=A+1}^{A+B} p_j^2 T_{ji}^2)\right) \quad (46)$$

$$\xrightarrow{T_{in}^2 \neq \mathbf{I}, \ T_{out}^2 = T^{hard}}$$

$$= \max[p_1^2, ..., p_A^2] - \sum_{i=1}^{A} p_i^2 (\sum_{j=1}^{A} p_j^2 T_{ji}^2 + \sum_{j \in H_i} p_j^2).$$

Thus, we have:

$$\Delta E_{x_1} - \Delta E_{x_2} = \sum_{i=1}^{A} p_i^2 (\sum_{j=1}^{A} p_j^2 T_{ji}^2 + \sum_{j \in H_i} p_j^2) - \sum_{i=1}^{A} p_i^1 (\sum_{j=1}^{A} p_j^1 T_{ji}^1 + \frac{1}{A}\sum_{i=A+1}^{A+B} p_i^1)$$

$$= \sum_{i=1}^{A} p_i^1 (\sum_{j \in H_i} p_j^1 - \frac{1}{A}\sum_{i=A+1}^{A+B} p_i^1). \quad (47)$$

We note that the result aligns with eq. (38). Therefore, the presence of additional open-set noise does not affect the conclusion in the overfitted case.

# E  REVISITING EXISTING LNL METHODS WITH OPEN-SET NOISE

In the main paper, we provide both theoretical analyses and empirical studies for two cases of interest, comparing the impact of different types of open-set noise on the model's generalization performance. In this section, we further examine the performance of existing learning with noisy labels (LNL) methods, particularly the more prominent sample selection-based approaches, in handling the various open-set label noise scenarios previously discussed. First, in appendix E.1, we evaluate the effectiveness of integrating the open-set noise detection mechanism, discussed in section 4.2, into methods that were not originally designed to address open-set noise. We then conduct benchmark tests on two additional methods that explicitly account for open-set noise in appendix E.1.

## E.1  AUGMENTING EXISTING LNL METHODS WITH ENTROPY-BASED OPEN-SET NOISE DETECTION

In this section, we evaluate two representative learning with noisy labels (LNL) methods with well-maintained open-source implementations: SSR (Feng et al., 2022) and DivideMix (Li et al., 2020). Further details about these methods can be found in appendix E.2. Briefly, as standard sample selection methods, both approaches typically consist of a sample selection module and a model training module. Here, we retain the model training module and focus specifically on the sample selection module. We examine the following three variants:

- SSR/DivideMix: The original method.

- EntSel: Replaces the original sample selection module in SSR/DivideMix with the open-set noise detection method discussed in section 4.2. For details on how samples are selected using the open-set noise detection method, please refer to appendix E.2.

- SSR/DivideMix + EntSel: Selects the intersection of samples chosen by both the open-set noise detection method and the original sample selection module.

Based on the theoretical analysis in section 4.2, we have the following expectations:

1. We **expect** EntSel to improve OOD detection performance, particularly for easy open-set noise, though it may result in reduced closed-set detection performance;

2. Since SSR/DivideMix + EntSel integrates two sample selection mechanisms, we **expect** improvements in both OOD detection and closed-set classification performance.

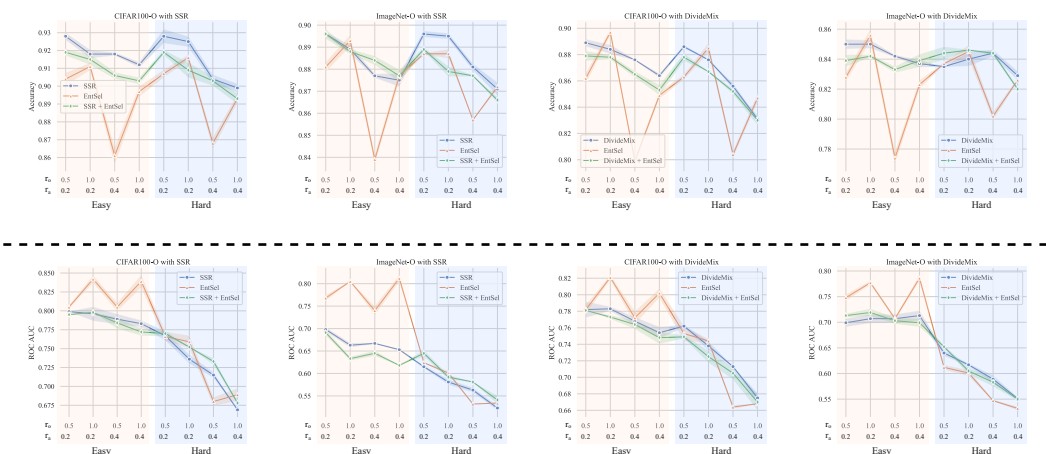

Figure 5: Evaluation of directly supervised training with different noise modes/ratios. First row: Closed-set classification accuracy; Second row: OOD detection ROC AUC.

In fig. 5, we present empirical results on CIFAR100-O and ImageNet-O with varying levels of open-set noise, as well as mixed noise scenarios that include both closed-set and open-set noise. First, we have confirmed that EntSel improves OOD detection performance while decreasing closed-set classification performance compared to the original methods. This effect is particularly pronounced when dealing with 'easy' open-set noise. However, SSR/DivideMix + EntSel does not enhance performance as anticipated. Upon further analysis, we observe that SSR/DivideMix + EntSel selects a significantly smaller subset of samples compared to either SSR/DivideMix or EntSel alone, likely due to the intersection of the selected samples. This suggests that the precision-recall trade-off in sample selection may be responsible for the performance decline. While combining both methods increases precision, it reduces recall, potentially eliminating noisy samples but also discarding clean ones. This indicates that using the intersection strategy may not be optimal. Effectively integrating open-set noise detection mechanisms with existing sample selection methods remains a promising area for future research.

**Results on real-world noisy dataset** We also present the results on the real-world WebVision dataset in table 1. Consistent with previous experiments on synthetic datasets, we observe similar trends between the two SSR method variants combined with EntSel and the original version. Specifically, EntSel (SSR) enhances OOD detection performance while reducing closed-set classification performance. Both SSR+EntSel and DivideMix+EntSel result in declines in classification accuracy and OOD detection performance. Notably, EntSel (DivideMix) does not improve OOD detection performance. Additional experiments reveal that EntSel is highly sensitive to hyperparameter tuning. For example, reducing the threshold $\theta'$ for EntSel significantly improves performance on the WebVision dataset, particularly when EntSel is integrated with DivideMix. Adjusting $\theta'$ from its default

value of 0.5 to 0.2 increases classification accuracy from 62.96% to 67.2% and raises the ROC AUC from 81.66% to 85.99%.

Interestingly, the original DivideMix, while achieving lower classification accuracy than the original SSR (table 1), achieves higher ROC AUC scores in OOD detection. This result suggests that classification accuracy alone may not provide a comprehensive evaluation of model performance—additional metrics, such as OOD detection, are necessary for a more complete assessment.

Table 1: Results on WebVision dataset.

| Method | Accuracy (%) | ROC AUC (%) |
|---|---|---|
| SSR | **77.48** | 80.84 |
| EntSel (SSR) | 77.08 | **85.43** |
| SSR + EntSel | 76.04 | 79.90 |
| DivideMix | **74.08** | **86.39** |
| EntSel (DivideMix) | 62.96 | 81.66 |
| DivideMix + EntSel | 58.94 | 83.85 |

**Benchmark more methods with newly-proposed open-set noise**   In this section, we present additional benchmarking results on the CIFAR100-O dataset across various open-set noise ratios and modes, as shown in table 2. To provide a more comprehensive analysis, we include two additional methods alongside SSR and DivideMix: EvidentialMix (Sachdeva et al., 2021) and DSOS (Albert et al., 2022), both of which propose tailored solutions for handling open-set noise during their design.

Table 2: Benchmarking results on CIFAR100-O datasets.

| Method / Noise Ratio | 0.2 Easy | 0.4 Easy | 0.2 Hard | 0.4 Hard |
|---|---|---|---|---|
| SSR (Feng et al., 2022) | 0.889 | 0.875 | 0.895 | 0.871 |
| DivideMix (Li et al., 2020) | 0.783 | 0.754 | 0.738 | 0.675 |
| EvidentialMix (Sachdeva et al., 2021) | 0.884 | 0.827 | 0.898 | 0.872 |
| DSOS (Albert et al., 2022) | 0.846 | 0.765 | 0.854 | 0.832 |

The results indicate that the various methods exhibit differing sensitivities to open-set noise. Notably, the tailored solutions for open-set noise in EvidentialMix and DSOS do not yield consistent improvements compared to standard methods like SSR. We acknowledge that this may be partially due to insufficient hyperparameter tuning. Nevertheless, the performance analysis is complex and warrants further investigation, as these methods typically involve multiple components and regularization strategies, which are beyond the scope of this paper.

E.2   DETAILS OF INVOLVED METHODS

**DivideMix**   (Li et al., 2020) Denoting as $\mathcal{L} = \{l_i\}_{i=1}^N$ the losses of all samples, DivideMix proposes to model it (after min-max normalization) with a Gaussian Mixture Model. The probabilities $\{p_i\}_{i=1}^N$ of each sample belonging to the component with the smaller mean value are then extracted. Samples with probability $p_i$ greater than the threshold $\theta$ are then identified as the "clean" subset. Link to code: https://github.com/LiJunnan1992/DivideMix.

**SSR**   (Feng et al., 2022) In contrast to DivideMix, SSR extracts features for each sample and constructs a neighbourhood graph. By computing the nearest neighbour labels for each sample, a pseudo-label distribution $\boldsymbol{p}$ is obtained through a kNN voting process. The consistency $c = \boldsymbol{p}_y/\boldsymbol{p}_{max}$ between this voted distribution and the given noisy label $y$ (logit label) is then calculated. Samples with consistency $c$ greater than the threshold $\theta$ are identified as part of the "clean" subset. Link to code: https://github.com/MrChenFeng/SSR_BMVC2022.

**EvidentialMix**   (Sachdeva et al., 2021) EvidentialMix adopts a structure fundamentally similar to DivideMix, but unlike DivideMix, which relies on cross-entropy loss for sample selection, it introduces

subjective logic loss as the selection criterion. This approach is believed to better differentiate open-set noise samples. Link to code: `https://github.com/ragavsachdeva/EvidentialMix`.

**DSOS** (Albert et al., 2022) DSOS also modifies the sample selection criteria. They propose a method called collision entropy, which can simultaneously identify both open-set and closed-set noise. Link to code: `https://github.com/PaulAlbert31/DSOS`.

**EntSel** We also provide a concise overview of the steps involved in EntSel. Denoting as $\mathcal{E} = \{e_i\}_{i=1}^N$ the entropy of all samples' predictions, we similarly model it (after min-max normalization) with a Gaussian Mixture Model. The probabilities $\{p_i\}_{i=1}^N$ of each sample belonging to the component with a smaller mean value are then extracted. Samples with probability $p_i$ greater than the threshold $\theta'$ are then identified as "inlier" subset and used for training.

## F ROBUST LOSS FUNCTIONS MEET OPEN-SET NOISE

We are happy to include more results of methods based on robust loss functions. Specifically, we considered some widely used robust loss functions, including the Symmetric Cross Entropy (SCE) loss function (Wang et al., 2019) and the Generalized Cross Entropy (GCE) loss function (Zhang and Sabuncu, 2018). We report below the experimental results (Classification accuracy and OOD detection AUC score) on the CIFAR100-O and ImageNet-O datasets after replacing the standard cross-entropy loss with two different robust loss functions.

Table 3: Classification accuracy with robust loss functions on CIFAR100-O dataset.

| Noise mode | Easy | | | | Hard | | | |
|---|---|---|---|---|---|---|---|---|
| Noise ratio | 0.1 | 0.2 | 0.3 | 0.4 | 0.1 | 0.2 | 0.3 | 0.4 |
| CE | 0.846 | 0.804 | 0.770 | 0.714 | **0.872** | 0.847 | **0.842** | **0.829** |
| GCE | **0.854** | 0.810 | 0.763 | 0.708 | 0.864 | 0.840 | 0.813 | 0.800 |
| SCE | 0.846 | **0.822** | **0.787** | **0.729** | 0.871 | **0.854** | 0.840 | 0.814 |

Table 4: Classification accuracy with robust loss functions on CIFAR100-O dataset.

| Noise mode | Easy | | | | Hard | | | |
|---|---|---|---|---|---|---|---|---|
| Noise ratio | 0.1 | 0.2 | 0.3 | 0.4 | 0.1 | 0.2 | 0.3 | 0.4 |
| CE | **0.804** | 0.793 | 0.773 | 0.754 | **0.770** | **0.728** | **0.692** | **0.664** |
| GCE | 0.782 | 0.771 | 0.752 | 0.719 | 0.759 | 0.718 | 0.679 | 0.639 |
| SCE | 0.794 | **0.799** | **0.784** | **0.756** | 0.749 | 0.718 | 0.682 | 0.651 |

Table 5: Classification accuracy with robust loss functions on ImageNet-O dataset.

| Noise mode | Easy | | | | Hard | | | |
|---|---|---|---|---|---|---|---|---|
| Noise ratio | 0.1 | 0.2 | 0.3 | 0.4 | 0.1 | 0.2 | 0.3 | 0.4 |
| CE | 0.822 | 0.783 | 0.752 | **0.721** | **0.859** | 0.838 | **0.834** | 0.821 |
| GCE | 0.813 | 0.788 | 0.739 | 0.714 | 0.853 | 0.833 | 0.818 | **0.834** |
| SCE | **0.826** | **0.797** | **0.759** | 0.720 | 0.841 | **0.839** | 0.831 | 0.827 |

We highlight the methods that achieve the best performance under different settings in bold. Overall, we observe the following: - Compared to the original CE loss, the GCE loss function generally results in lower classification accuracy and OOD detection AUC scores. - The SCE loss function appears to improve the classification and OOD detection performance in the presence of 'Easy' open-set noise. However, it seems to degrade performance when dealing with 'Hard' open-set noise.

Table 6: Classification accuracy with robust loss functions on ImageNet-O dataset.

| Noise mode | Easy | | | | Hard | | | |
|---|---|---|---|---|---|---|---|---|
| Noise ratio | 0.1 | 0.2 | 0.3 | 0.4 | 0.1 | 0.2 | 0.3 | 0.4 |
| CE | **0.769** | 0.760 | 0.764 | 0.739 | **0.658** | **0.601** | **0.569** | **0.549** |
| GCE | 0.732 | 0.740 | 0.729 | 0.719 | 0.636 | 0.591 | 0.555 | 0.513 |
| SCE | 0.749 | **0.768** | **0.765** | **0.748** | 0.633 | 0.599 | 0.558 | 0.537 |

Nevertheless, we want to emphasize that the performance differences between the two robust loss functions and the original cross-entropy loss in the above results are not significant. Furthermore, these robust loss functions were not originally designed to account for open-set noise. Therefore, we believe further analysis is needed to evaluate the performance of different robust loss functions under open-set noise, and we leave it to our future work.

That said, we would like to offer some preliminary insights. We want to point out that these robust loss functions generally only affect the convergence speed but do not alter the fully converged extrema. For instance, in the case of the Symmetric Cross-Entropy (SCE) loss, we have:

$$\mathcal{L}_{\text{SCE}} = \alpha \cdot \mathcal{L}_{\text{CE}} + \beta \cdot \mathcal{L}_{\text{RCE}}$$

where: $\mathcal{L}_{\text{CE}} = -\sum_{i=1}^{C} y_i \log p_i$, $\mathcal{L}_{\text{RCE}} = -\sum_{i=1}^{C} p_i \log y_i$, $\alpha$ and $\beta$ are weighting coefficients for the two terms.

$$\frac{\partial \mathcal{L}_{\text{SCE}}}{\partial z_i} = \alpha \cdot \frac{\partial \mathcal{L}_{\text{CE}}}{\partial z_i} + \beta \cdot \frac{\partial \mathcal{L}_{\text{RCE}}}{\partial z_i}$$

Breaking it down:

- Gradient of CE Term: $\frac{\partial \mathcal{L}_{\text{CE}}}{\partial z_i} = p_i - y_i$
- Gradient of RCE Term: $\frac{\partial \mathcal{L}_{\text{RCE}}}{\partial z_i} = \frac{y_i}{p_i} \cdot (1 - p_i)$
- Gradient of SCE Loss: $\frac{\partial \mathcal{L}_{\text{SCE}}}{\partial z_i} = \alpha \cdot (p_i - y_i) + \beta \cdot \frac{y_i}{p_i} \cdot (1 - p_i)$

For the true class ($i = y$): $\frac{\partial \mathcal{L}_{\text{SCE}}}{\partial z_y} = \alpha \cdot (p_y - 1) + \beta \cdot \frac{1}{p_y} \cdot (1 - p_y)$

For all other classes ($i \neq y$): $\frac{\partial \mathcal{L}_{\text{SCE}}}{\partial z_i} = \alpha \cdot p_i + \beta \cdot \frac{0}{p_i} \cdot (1 - p_i) = \alpha \cdot p_i$

We notice that, for both CE loss and SCE loss, their gradients reduce to 0 if and only if $p_i = y_i, \forall i$, which corresponds to the **overfitted case** analyzed in our paper. This implies that with sufficient model capacity and training (as is often the case with modern deep neural networks), the conclusions of our analysis remain valid even when robust loss functions are used.

## G PRELIMINARY EXPLORATIONS ON HANDLING DIFFERENT OPEN-SET NOISE SCENARIOS

While we would like to reiterate that our aim in this work is not to propose a new empirical solution, we are happy to provide some potential ideas. Based on the theoretical analysis presented in our paper, we observe that existing methods, such as entropy-based detection mechanisms, may struggle to handle 'hard' open-set noise—this type of noise primarily arises from semantic similarities between open-set noise and closed-set categories. Below, we explore two different methods and present the results of preliminary experiments.

**Entropy-based open-set noise detection with trained encoder** We first investigate whether pretrained encoders can assist in identifying open-set noise. Compared to randomly initialized feature spaces, we expect that pretrained encoders, with their better-organized representations, may more

effectively distinguish challenging open-set samples. Specifically, we observe the entropy dynamics of open-set noise and clean samples after replacing the randomly initialized encoder in the main paper with a pretrained encoder.

We first consider **self-supervised pretraining**. Specifically, we apply the MoCo framework (He et al., 2020) to pretrain the encoder for 500 epochs. Below, we show the entropy dynamics at different warmup training epochs with pretrained encoder:

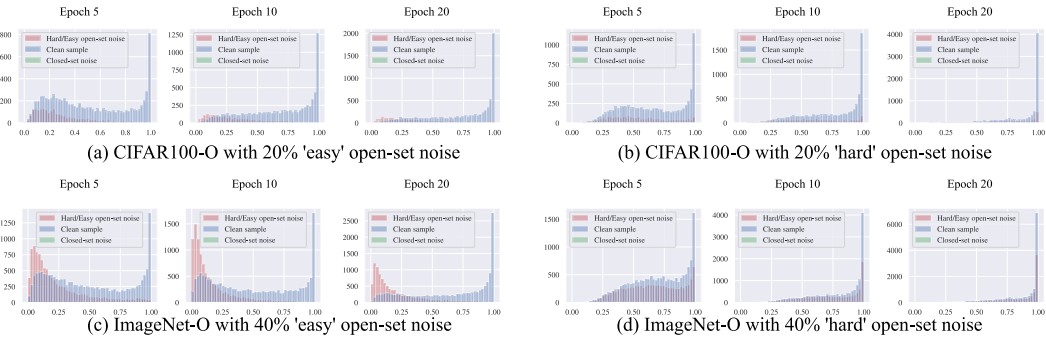

Figure 6: Entropy dynamics with Self-supervised pretrained encoder.

We also consider to utilize the **pretrained vision encoder of CLIP model** (Radford et al., 2021).

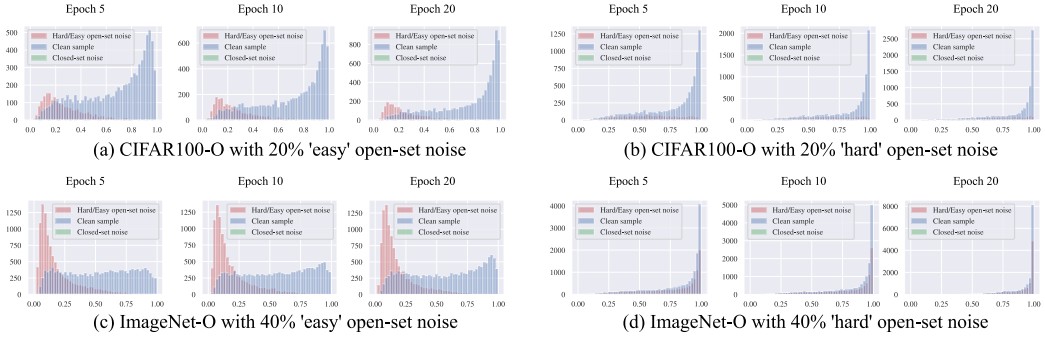

Figure 7: Entropy dynamics with CLIP encoder (VIT-B/32).

Unfortunately, by comparing fig. 6 and fig. 7 above with fig. 3 in the paper, we observe that neither of the two pretrained encoders results in noticeable improvements. The entropy-based open-set noise detection mechanisms remain effective only for 'easy' open-set noise and continue to show insensitivity to 'hard' open-set noise.

**Zeroshot open-set noise detection with CLIP** Due to its multi-modality nature, we further try to utilize CLIP for zero-shot open-set noise detection. Specifically, we design a simple algorithm to compute an intuitive indicator value for identifying open-set noise. For each sample $x$ with annotated label $y$,

*1. Generate Text Prompts:* For the target class $y$, we create a text prmopt: "A photo of class $y$.". For non-target classes, we consider a set of prompts: ["A photo of class $i$." for $i \in L_y$]. Here, we denote as $L_y$ the possible source classes to which the sample $x$ may belong. Practically, $L_y$ can be a broad set of classes, such as the 1K classes from ImageNet-1K dataset, or it can be manully defined to include semanticlly-challenging classes; for example, ['tiger', 'cheetah'] for class 'cat'. In below experiments, we default to the first option.

*2. Calculate Similarities:* We first compute similarity to the target class: $S_y = \text{sim}(v_x, t_y)$. Here, $v_x$ and $t_y$ denotes the visual and textual representation, respetively. We also compute a maximum similarities to non-target classes: $S_{\text{other}} = \max\{\text{sim}(v_x, t_i) \mid i \in L_y\}$.

*3. Compute the Difference:* $D_x = S_y - S_{\text{other}}$.

Intuitively, we measure and compare the similarity of the visual semantics of sample $x$ to its annotated text label and the most likely labels from the source classes. To illustrate the effectiveness of $D_x$ as an open-set noise indicator, we plot the distribution of $D_x$ for different samples below:

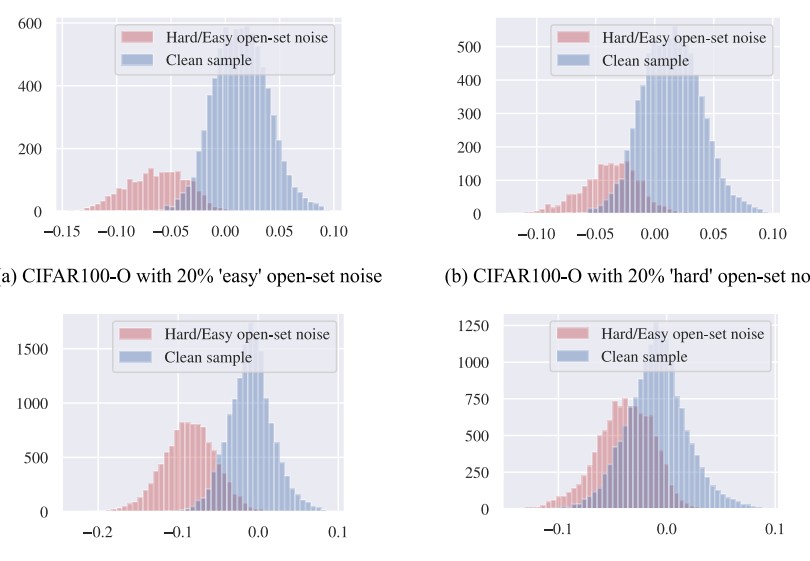

(a) CIFAR100-O with 20% 'easy' open-set noise     (b) CIFAR100-O with 20% 'hard' open-set noise

(c) ImageNet-O with 40% 'easy' open-set noise     (d) ImageNet-O with 40% 'hard' open-set noise

Figure 8: Zeroshot open-set noise detection with CLIP.

We notice that, compared to the entropy-based open-set detection mechanism, the zero-shot open-set identification brings steady improvements.

## H    MORE EXAMPLES OF OPEN-SET NOISE IN WEBVISION DATASET

In this section, we provide additional examples of open-set noise within the 'Tench' class of the WebVision dataset. By tracing the origins of web pages hosting these open-set noise images, we found that the term "Tench" or related keywords frequently appeared on these pages. We attribute this to the data collection process on the web, where images were inadvertently included during keyword searches and web crawling due to the presence of relevant keywords in image descriptions or surrounding text, such as people with "Tench" in their name or related fishing tools.

As noted earlier, the prevailing belief in the current LNL community is that real-world noise predominantly stems from semantic similarity. Consequently, recent research has largely focused on instance-dependent noise and its theoretical analysis. However, our findings suggest that, in real-world scenarios—particularly in web-crawled datasets—noise may not always be linked to semantics, but may also arise from latent high-dimensional factors, such as accompanying text. Addressing different types of real-world noise also warrants increased attention and further exploration.

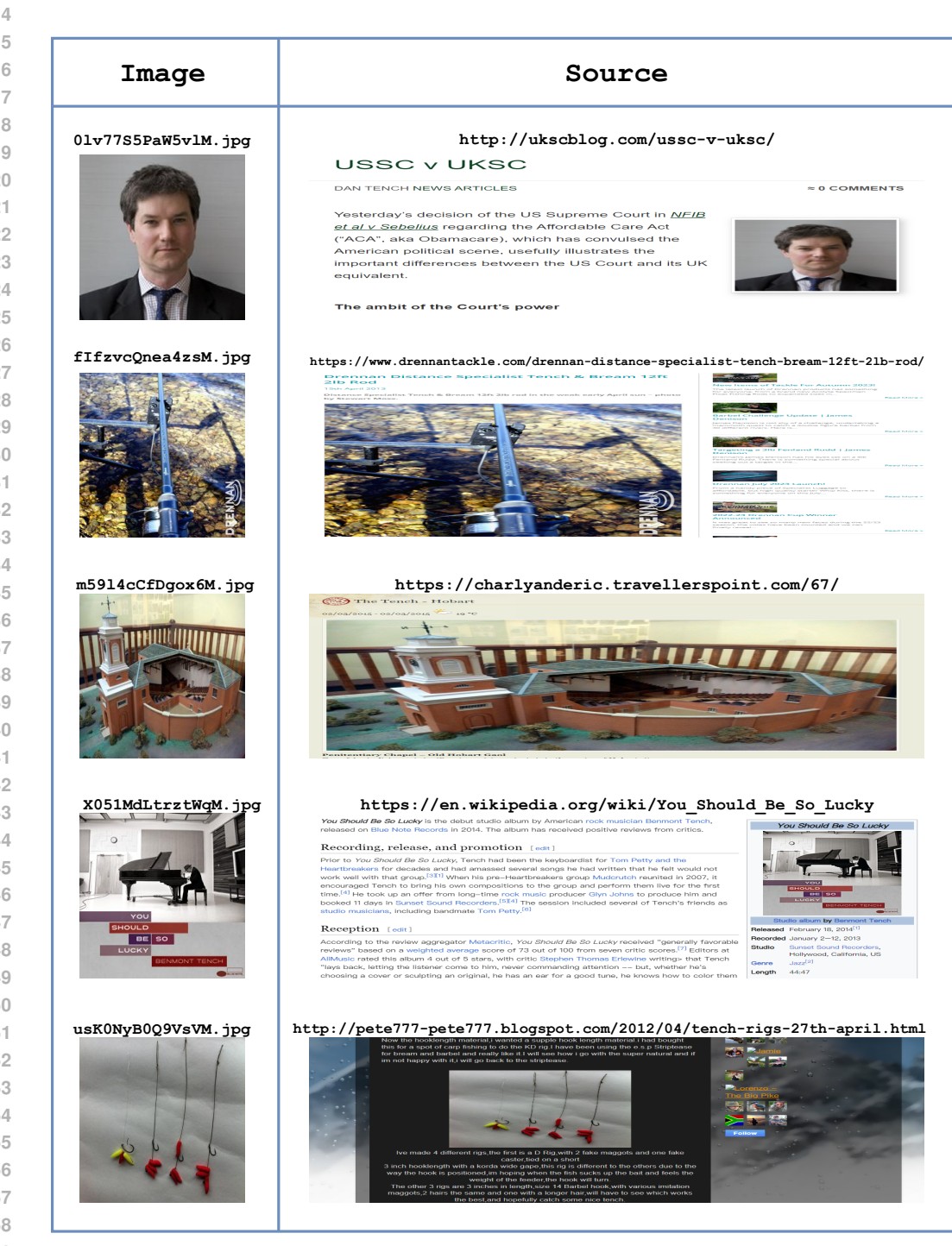

Figure 9: Open-set noise examples in class "Tench" of WebVision dataset with path: /google/q0001/. The source images are resized to fit the layout. Please note that the web links here are obtained in May 2024 and validated effective in Sept 2024, and there is no guarantee that they will always be valid in the future.

