# OpenReview forum: "Towards Better Understanding Open-set Noise in Learning with Noisy Labels"
_ICLR.cc/2025/Conference — Submitted to ICLR 2025_

### Official Review · Reviewer_rg5p · 2024-10-27

**Soundness:** 3
**Presentation:** 3
**Contribution:** 2
**Rating:** 5
**Confidence:** 4

**Summary:**

In this paper, the authors theoretically analyze and compare the impacts of open-set and closed-set noise, as well as the differences between various open-set noise modes. Additionally, they examine a common open-set noise detection mechanism based on
prediction entropy. Moreover, to validate their insights, they construct two open-set noisy datasets and a open-set test-set for evaluation.

**Strengths:**

1.The paper is easy to follow and the conclusions are easy to understand.

2.The mathematical analysis is adequate and there are experimental results to support these insights.

3.The experimental figures are clear and adequate on the constructed dataset.

**Weaknesses:**

1.The way of constructing benchmark is quite similar to existing benchmarks in LNL. Thus it may be not suitable to list it as a contribution.

2.Though the author introduces a hard open-set noise (seems like a combination of feature-dependent noise and open-set noise), but it seems that the author does not design a method to tackle such kind of noise.

3.Some of the conclusions may be naive and simple.  For example, " it may be effective only for ‘easy’ open-set noise.". Because entropy-based methods generally fail to detect close-set feature-dependent noise as well.


There are some minor issues that will not affect the rating:
1." the concept of complete noise transition matrix" should be " the concept of a complete noise transition matrix".

2. "namely fitted case and overfitted case" should be "namely the fitted case and overfitted case".

3.It would be better if the contributions are more compact. There are six contributions now.

**Questions:**

What does this sentence mean: obtaining a model that perfectly fits the data distribution is often challenging; here, we consider training a single-layer linear classifier upon a frozen pretrained encoder. I think a single-layer linear classifier may even be worse to fit the data distribution.

---

> ### Author Response · Authors · 2024-11-23
> **Response to Reviewer rg5p (Part 1/3)**
>
> Thanks very much for the careful review. We sincerely appreciate your time and effort in reading our paper, as well as the insightful and constructive feedback. To fully address your questions, we have conducted additional experiments and analyses. We kindly ask for your understanding regarding the slightly lengthy response and sincerely look forward to engaging in further discussions with you.
>
> > **W1: Benchmark construction: *'The way of constructing benchmark is quite similar to existing benchmarks in LNL.'***
>
> Many thanks for the comment. Regarding *'the way of constructing benchmark'*, we would like to reiterate the content from our paper to emphasize once again how our approach to dataset construction differs from existing methods:
>
> *`Previous works involving open-set noise also attempted to build synthetic noisy datasets, typically treating different datasets as open-set noise for each other to construct synthetic noisy dataset (Sachdeva et al., 2021; Wu et al., 2021). In this scenario, potential domain gaps could affect a focused analysis of open-set noise. In this work, we propose selecting inlier/outlier classes from the same dataset to avoid this issue. Besides, in previous works, the consideration of open-set noise modes often focused on random flipping from outlier classes to all possible inlier classes, which is indeed the ‘easy’ open-set noise adopted in this paper.`*
>
> We emphasize the following key points:
> 1. **Avoiding domain gaps**: Our construction method selects inlier and outlier classes from the same dataset, minimizing the implicit effects of domain gaps in previous works.
> 2. **Addressing both 'easy' and 'hard' open-set noise**: Unlike prior benchmarks, we consider multiple open-set noise modes, including the challenging 'hard' open-set noise.
>
> **We understand that the reviewer might prefer a newly collected, large-scale open-set noisy dataset. Regrettably, this was not the goal of our work. We consider such a contribution to be more suitable as a dataset & benchmark track paper, and the budget required to collect such a dataset is beyond our current resources.** Nonetheless, we made an extra effort to provide an open-set test dataset based on the real-world noisy dataset, WebVision. We hope this additional contribution addresses the reviewer’s concerns and demonstrates our commitment to advancing the study of open-set noise in a meaningful way.
>
> > **W2: Method for learning with hard open-set noise: *'... it seems that the author does not design a method to tackle such kind of noise.'***
>
> Many thanks for the question. While we would like to reiterate that our aim in this work is not to propose a new empirical solution, we are happy to provide some potential ideas. Based on the theoretical analysis presented in our paper, we observe that existing methods, such as entropy-based detection mechanisms, may struggle to handle 'hard' open-set noise—this type of noise primarily arises from semantic similarities between open-set noise and closed-set categories. **Below, we explore two different methods and present the results of preliminary experiments.**
>
> **1. Entropy-based open-set noise detection with trained encoder**
>
> We first investigate whether pretrained encoders can assist in identifying open-set noise. Compared to randomly initialized feature spaces, we expect that pretrained encoders, with their better-organized representations, may more effectively distinguish challenging open-set samples. Specifically, we observe the entropy dynamics of open-set noise and clean samples after replacing the randomly initialized encoder in the main paper with a pretrained encoder.
>
> We first consider **self-supervised pretraining**. Specifically, we apply the MoCo framework [1] to pretrain the encoder for 500 epochs. Below, we show the entropy dynamics at different warmup training epochs with pretrained encoder:
>
> *Figure R1. Entropy dynamics with Self-supervised pretrained encoder* [(Clickable anonymous link)](https://anonymous.4open.science/r/ICLR25-submission4732/SelfSupPretrained_entropydynamic.pdf)
>
> We also consider to utilize the **pretrained vision encoder of CLIP model** [2].
>
> *Figure R2. Entropy dynamics with CLIP encoder (VIT-B/32)* [(Clickable anonymous link)](https://anonymous.4open.science/r/ICLR25-submission4732/CLIPencoder_entropydynamic.pdf)
>
>
> **Unfortunately, by comparing Figures R1/2 above with Figure 3 in the paper, we observe that neither of the two pretrained encoders results in noticeable improvements. The entropy-based open-set noise detection mechanisms remain effective only for 'easy' open-set noise and continue to show insensitivity to 'hard' open-set noise.**

---

> > ### Author Response · Authors · 2024-11-28
> > **Invitation for further discussion**
> >
> > Dear Reviewer rg5p,
> >
> > Thank you very much for your detailed review and valuable feedback on our submission. **Your insights are crucial for us to optimize and improve the quality of our work.** We have carefully considered your suggestions and have made several revisions to the manuscript.
> >
> > **In particular, based on the theoretical insights provided in the paper, we have considered two additional methods to address the challenges of 'hard' open-set noise and conducted preliminary experimental explorations.** We have found that leveraging vision-language models to address open-set noise holds great potential, and we sincerely hope these explorations align with your expectations.
> >
> > Additionally, based on the suggestions from *Reviewer op1H*, **we have also examined the performance of robust loss functions in handling open-set noise**. We sincerely hope this addition can further earn your support.
> >
> > **If you have any additional suggestions or comments, we would greatly appreciate the opportunity for us to provide further clarifications before your final decision.** We believe that your guidance would significantly enhance the quality of our manuscript.
> >
> > Thank you once again for your time and support. We truly appreciate your help and look forward to your response.
> >
> > Best regards,
> >
> > Authors of Submission 4732

---

> ### Author Response · Authors · 2024-11-23
> **Response to Reviewer rg5p (Part 2/3)**
>
> **2. Zeroshot open-set noise detection with CLIP**
>
> Due to its multi-modality nature, we further try to utilize CLIP for zero-shot open-set noise detection. Specifically, we design a simple algorithm to compute an intuitive indicator value for identifying open-set noise. For each sample $x$ with annotated label $y$,
> 1. Generate Text Prompts:
>    - For the target class $y$, we create a text prmopt: "A photo of class {$y$}.".
>    - For non-target classes, we consider a set of prompts: ["A photo of class {$i$}." for $i$ $\in$ $L_y$]. Here, we denote as $L_y$ the possible source classes to which the sample $x$ may belong. Practically, $L_y$ can be a broad set of classes, such as the 1K classes from ImageNet-1K dataset, or it can be manully defined to include semanticlly-challenging classes; for example, ['tiger', 'cheetah'] for class 'cat'. In below experiments, we default to the first option.
>
> 2. Calculate Similarities:
>    - Similarity to the target class:
>      $S_y = \text{sim}(v_x, t_y)$. Here, $v_x$ and $t_y$ denotes the visual and textual representation, respetively.
>    - Maximum similarities to non-target classes:
>      $S_{\text{other}} = \max \{ \text{sim}(v_x, t_{i}) \mid i \in L_y \}$.
>
> 3. Compute the Difference: $D_x = S_y - S_{\text{other}}$.
>
> Intuitively, we measure and compare the similarity of the visual semantics of sample $x$ to its annotated text label and the most likely labels from the source classes.  To illustrate the effectiveness of $D_x$ as an open-set noise indicator, we plot the distribution of $D_x$ for different samples below：
>
> *Figure R3. Zeroshot open-set noise detection with CLIP* [(Clickable anonymous link)](https://anonymous.4open.science/r/ICLR25-submission4732/CLIP_zeroshot_selection.pdf)
>
> **We notice that, compared to the entropy-based open-set detection mechanism, the zero-shot open-set identification brings steady improvements.**
>
> Please note that these are preliminary experiments and initial attempts—we plan to explore this more deeply in our future work. We have also updated the manuscript `Appendix G` with the new results mentioned above. We hope this new analysis helps address your concerns.
>
> *[1] He, Kaiming, et al. "Momentum contrast for unsupervised visual representation learning." Proceedings of the IEEE/CVF conference on computer vision and pattern recognition. 2020.*
>
> *[2] Radford, Alec, et al. "Learning transferable visual models from natural language supervision." International conference on machine learning. PMLR, 2021.*
>
> > **W3: Significance of conclusions: *'Some of the conclusions may be naive and simple. For example, " it may be effective only for ‘easy’ open-set noise.". Because entropy-based methods generally fail to detect close-set feature-dependent noise as well.'***
>
> Many thanks for the comment. **We fully understand the insights from closed-set noise to open-set noise—indeed, this was one of the baseline assumptions we considered prior to conducting our analysis and experiments. Nevertheless, we believe that rigorous theoretical analysis and experimental validation are necessary.** Through these efforts, we not only confirmed the validity of the assumption but also revealed the limitations of existing methods in addressing challenging 'hard' open-set noise.
>
> To summarize, we hope our rigorous analysis and clear conclusions will inspire further research on open-set noise, moving beyond the traditional focus on closed-set noise and entropy-based methods. **Additionally, based on **Reviewer op1H**'s suggestion, we have included additional analysis in the updated appendix, including the performance check of robust loss functions in dealing with different open-set noise, to further clarify and support our findings.** We sincerely hope these supplements can further address your concerns.
>
>
> > **Q1: Linear classifier for fitted case: *'What does this sentence mean: obtaining a model that perfectly fits the data distribution is often challenging; here, we consider training a single-layer linear classifier upon a frozen pretrained encoder. I think a single-layer linear classifier may even be worse to fit the data distribution.'***
>
> Many thanks for the question. We are happy to further clarify: by 'perfectly fits the data distribution', we mean the model has perfectly fit the sampled distribution, i.e., the *fitted case*.
>
> It is known that there is always a trade-off between underfitting and overfitting with respect to model capacity. A single-layer linear layer has limited capacity, thus it is harder to memorize all the labels, i.e., overfitting to the training set. However, as you mentioned, a single-layer linear classifier may also struggle to fully fit the data distribution due to its limited capacity. Thus, we propose to start with a frozen pretrained encoder - which provides a well-learned representation feature space. This approach reduces the burden on the linear classifier and allows it to leverage the pretrained features for classification.

---

> ### Author Response · Authors · 2024-11-23
> **Response to Reviewer rg5p (Part 3/3)**
>
> > **Minor issues on presentation.**
>
> Many thanks for your careful reading. We would like to express our deep gratitude for your suggestions on the presentation. **We have proofread the paper again and updated the manuscript accordingly based on your suggestions (we kindly refer you to the updated paper).** Specifically, the contributions are condensed into three points as below (Following your suggestion, we removed the dataset contribution and incorporated additional analysis and experiments):
> - *`We introduce the concept of a complete noise transition matrix, reformulate the Learning with Noisy Labels (LNL) problem to account for open-set noise, and analyze two offline cases: fitted and overfitted.`*
> - *`We demonstrate that open-set noise generally has less negative impact on classification accuracy than closed-set noise, analyze 'hard' vs. 'easy' open-set noise, propose an out-of-distribution (OOD) detection task for further evaluation, and find entropy-based open-set noise detection effective only for 'easy' open-set noise.`*
> - *`We conduct preliminary explorations with vision-language models and self-supervised models on identifying and learning with 'hard' open-set noise, expand experiments on the performance of robust loss functions under open-set noise, and analyze their effectiveness in challenging noise scenarios.`*
>
>
> **Thank you once again for your time and valuable feedback, which have helped us improve our work. If there are any additional questions or clarifications needed, please do not hesitate to let us know. We would also kindly ask if you might consider revising your score, should our responses satisfactorily resolve the points you raised. We truly appreciate your consideration and look forward to your further comments.**

---

> ### Author Response · Authors · 2024-11-30
>
> Dear Reviewer rg5p,
>
> We sincerely apologize for reaching out to you again, and we hope this does not disrupt your Thanksgiving holiday if you celebrate it. **As the discussion phase will conclude in two days, we wonder if you have any additional suggestions or concerns. We would be more than happy to engage in further discussions. If our rebuttal has helped to resolve your concerns, we sincerely hope you might consider adjusting the rating—your support is crucial for this borderline submission.**
>
> Thank you once again for your time and support. We greatly appreciate your detailed review and valuable feedback on our submission. Your insights have been instrumental in helping us refine and improve the quality of our work. We are truly grateful for your assistance and look forward to your reply.
>
> Best regards,
>
> Authors of Submission 4732

---

### Official Review · Reviewer_FqR8 · 2024-11-02

**Soundness:** 2
**Presentation:** 2
**Contribution:** 2
**Rating:** 3
**Confidence:** 3

**Summary:**

This paper analyzes the learning with noisy labels problem by explicitly accounting for the presence of open-set noise. However, only theoretical analysis and dataset construction are not sufficient to be published in ICLR.

**Strengths:**

This paper analyzes the learning with noisy labels problem by explicitly accounting for the presence of open-set noise.

**Weaknesses:**

Only theoretical analysis and dataset construction are not sufficient to be published in ICLR.

**Questions:**

No

---

> ### Author Response · Authors · 2024-11-23
> **Response to Reviewer FqR8**
>
> We regret that the feedback provided was brief and lacked depth. Based on the comments, it appears that the reviewer may not have fully engaged with our paper. In addition to the theoretical analysis and the newly proposed benchmark datasets, **our work also includes extensive experimental validations of the theoretical findings**, specifically addressing the following points:
>
> - An empirical comparison between open-set noise and closed-set noise in both the fitted and overfitted cases.
> - An evaluation of different open-set noise patterns under the same total noise ratio, in both the fitted and overfitted cases.
> - An analysis of the sensitivity of a representative entropy-based open-set detection method across various open-set noise patterns.
>
> Motivated by constructive feedback from **Reviewer op1H** and **Reviewer rg5p**, we have conducted further analyses, including:
> - The performance of robust loss functions under open-set noise.
> - Several potential solutions to address the newly proposed open-set noise scenarios.
>
> Furthermore, we respectfully disagree with the assertion that theoretical analysis and dataset construction are insufficient for publication at ICLR. As in many other areas of research, **rigorous theoretical exploration and the introduction of novel datasets can provide valuable insights and advance the understanding within the machine learning community**. We believe our contributions meet these standards and hope our additional analysis further strengthens our case.

---

### Official Review · Reviewer_op1H · 2024-11-04

**Soundness:** 3
**Presentation:** 2
**Contribution:** 3
**Rating:** 6
**Confidence:** 4

**Summary:**

This paper addresses the challenge of open-set noise in learning with noisy labels (LNL), a problem where the true labels of noisy samples may not belong to any known category. The authors refine the LNL problem by accounting for open-set noise and theoretically analyze its impact compared to closed-set noise. They construct two open-set noisy datasets and introduce a novel open-set test set for the WebVision dataset to empirically validate their findings. The results indicate that open-set noise has distinct characteristics and a lesser negative impact on model performance compared to closed-set noise, highlighting the need for further exploration into model evaluation under such conditions. The paper also examines an entropy-based open-set noise detection mechanism and proposes additional out-of-distribution detection tasks for model evaluation.

**Strengths:**

- The paper studies a detailed examination of open-set noise in the context of learning with noisy labels, an area that has been largely overlooked in previous research.

- It offers a robust theoretical framework to analyze the effects of open-set noise and supports these findings with empirical evidence through the creation and testing on synthetic datasets.

- The paper provides a careful look at different modes of open-set noise, comparing 'easy' and 'hard' open-set noise scenarios, which is crucial for understanding how various types of noise affect model performance.

- It introduces the use of out-of-distribution (OOD) detection as a complementary evaluation metric to traditional accuracy measures, enhancing the assessment of model performance in the presence of open-set noise.

- The paper underscores the significance of open-set noise in real-world datasets and demonstrates the practical impact of its findings on existing learning methods, highlighting the need for more research in this area.

**Weaknesses:**

- The paper examines a few existing learning with noisy labels (LNL) methods on the synthetic datasets, outlined in Appendix E.1. However, it does not explore a wide range of existing methods such as robust losses in LNL, which could limit the comprehensiveness of the comparison and the conclusions drawn about the state-of-the-art in handling open-set noise.

- For Section 3.3.2, Authors exclude the effect of closed-set noise (Cx = 0) and only focus on open-set noise which could limits the findings in real-world scenarios. For example, It is not clear how different open-set noise with same close-set label noise.

- Experiments on the main paper are not thorough enough. I suggest add more LNL methods and discusses how different methods behave and align with the theorems in the main paper.

**Questions:**

- For Figure 2 (c) (d), Authors state in the paper that "the presence of open-set noise degrades OOD detection performance, whereas, conversely, the presence of closed-set noise could even improve OOD detection performance.". However, from my observation, the closed-set noise does not improve detection performance from the Figure.

- In Section 3.3.2, Is it possible to assume the same pattern and noise ratio of close-set noise and then study how different open-set noise compare to each other?

- For Figure 2, is it drawn when model converges for each case?

---

> ### Author Response · Authors · 2024-11-23
> **Response to Reviewer op1H (Part 1/4)**
>
> Thanks very much for the careful review. We sincerely appreciate your time and effort in reading our paper, as well as the insightful and positive feedback. To fully address your questions, we have conducted additional experiments and analyses. We kindly ask for your understanding regarding the slightly lengthy response and sincerely look forward to engaging in further discussions with you.
>
> > **W1: Results on more LNL methods: *‘The paper examines a few existing learning with noisy labels (LNL) methods … the state-of-the-art in handling open-set noise.'***
>
> Many thanks for the suggestion. We mainly experimented with sample selection methods before, as these methods often hold the state-of-the-art performances on current benchmarks. We are happy to include more results of methods based on robust loss functions. Specifically, we considered some widely used robust loss functions, including the Generalized Cross Entropy (GCE) loss function [1] and the Symmetric Cross Entropy (SCE) loss function [2]. We report below the experimental results (Classification accuracy and OOD detection AUC score) on the CIFAR100-O and ImageNet-O datasets after replacing the standard cross-entropy loss with two different robust loss functions.
>
> | Noise mode      | Easy    |         |         |         | Hard    |         |         |         |
> |-----------------|---------|---------|---------|---------|---------|---------|---------|---------|
> | **Noise ratio** | **0.1** | **0.2** | **0.3** | **0.4** | **0.1** | **0.2** | **0.3** | **0.4** |
> | CE              | 0.846   | 0.804   | 0.770   | 0.714   | **0.872**   | 0.847   | **0.842**   | **0.829**   |
> | GCE [1]         | **0.854**   | 0.810   | 0.763   | 0.708   | 0.864   | 0.840   | 0.813   | 0.800   |
> | SCE [2]         | 0.846   | **0.822**   | **0.787**   | **0.729**   | 0.871   | **0.854**   | 0.840   | 0.814   |
>
> *Table 1. Classification accuracy with robust loss functions on CIFAR100-O dataset.*
>
>
> | Noise mode      | Easy    |         |         |         | Hard    |         |         |         |
> |-----------------|---------|---------|---------|---------|---------|---------|---------|---------|
> | **Noise ratio** | **0.1** | **0.2** | **0.3** | **0.4** | **0.1** | **0.2** | **0.3** | **0.4** |
> | CE              | **0.804**   | 0.793   | 0.773   | 0.754   | **0.770**   | **0.728**  | **0.692**   | **0.664**   |
> | GCE [1]         | 0.782   | 0.771   | 0.752   | 0.719   | 0.759   | 0.718   | 0.679   | 0.639   |
> | SCE [2]         | 0.794   | **0.799**  | **0.784**   | **0.756**   | 0.749   | 0.718   | 0.682   | 0.651   |
>
> *Table 2. OOD detection AUC score with robust loss functions on CIFAR100-O dataset.*
>
> | Noise mode      | Easy    |         |         |         | Hard    |         |         |         |
> |-----------------|---------|---------|---------|---------|---------|---------|---------|---------|
> | **Noise ratio** | **0.1** | **0.2** | **0.3** | **0.4** | **0.1** | **0.2** | **0.3** | **0.4** |
> | CE              | 0.822   | 0.783   | 0.752   | **0.721**   | **0.859**  | 0.838   | **0.834**   | 0.821   |
> | GCE [1]         | 0.813   | 0.788   | 0.739   | 0.714   | 0.853   | 0.833   | 0.818   | **0.834**   |
> | SCE [2]         | **0.826**   | **0.797**  | **0.759**   | 0.720   | 0.841   | **0.839**   | 0.831   | 0.827   |
>
> *Table 3. Classification accuracy with robust loss functions on ImageNet-O dataset.*
>
>
> | Noise mode      | Easy    |         |         |         | Hard    |         |         |         |
> |-----------------|---------|---------|---------|---------|---------|---------|---------|---------|
> | **Noise ratio** | **0.1** | **0.2** | **0.3** | **0.4** | **0.1** | **0.2** | **0.3** | **0.4** |
> | CE              | **0.769**   | 0.760   | 0.764   | 0.739   | **0.658**   | **0.601**   | **0.569**   | **0.549**   |
> | GCE [1]         | 0.732   | 0.740   | 0.729   | 0.719   | 0.636   | 0.591   | 0.555   | 0.513   |
> | SCE [2]         | 0.749   | **0.768**   | **0.765**   | **0.748**   | 0.633   | 0.599   | 0.558   | 0.537   |
>
> *Table 4. OOD detection AUC score with robust loss functions on ImageNet-O dataset.*
>
>
> We highlight the methods that achieve the best performance under different settings in bold. Overall, we observe the following:
> - Compared to the original CE loss, the GCE loss function generally results in lower classification accuracy and OOD detection AUC scores.
> - The SCE loss function appears to improve the classification and OOD detection performance in the presence of 'Easy' open-set noise. However, it seems to degrade performance when dealing with 'Hard' open-set noise.

---

> ### Author Response · Authors · 2024-11-23
> **Response to Reviewer op1H (Part 2/4)**
>
> Nevertheless, we want to emphasize that the performance differences between the two robust loss functions and the original cross-entropy loss in the above results are **not significant**. Furthermore, these robust loss functions were **not originally designed to account for open-set noise**. Therefore, we believe further analysis is needed to evaluate the performance of different robust loss functions under open-set noise, and we are very interested in exploring this in future work.
>
> That said, we would like to offer some preliminary insights. We want to point out that these robust loss functions generally only affect the convergence speed but do not alter the fully converged extrema. For instance, in the case of the Symmetric Cross-Entropy (SCE) loss, we have:
>
> $
> \mathcal{L}\_{\text{SCE}} = \alpha \cdot \mathcal{L}\_{\text{CE}} + \beta \cdot \mathcal{L}\_{\text{RCE}}
> $
>
> where:
> - $\mathcal{L}_{\text{CE}} = -\sum\_{i=1}^C y_i \log p_i$,
> - $\mathcal{L}_{\text{RCE}} = -\sum\_{i=1}^C p_i \log y_i$,
> - $\alpha$ and $\beta$ are weighting coefficients for the two terms.
>
> $
> \frac{\partial \mathcal{L}\_{\text{SCE}}}{\partial z_i} = \alpha \cdot \frac{\partial \mathcal{L}\_{\text{CE}}}{\partial z_i} + \beta \cdot \frac{\partial \mathcal{L}\_{\text{RCE}}}{\partial z_i}
> $
>
> Breaking it down:
>
> 1. Gradient of CE Term:
> $\frac{\partial \mathcal{L}_{\text{CE}}}{\partial z_i} = p_i - y_i$
>
> 2. Gradient of RCE Term:
> $
> \frac{\partial \mathcal{L}_{\text{RCE}}}{\partial z_i} = \frac{y_i}{p_i} \cdot (1 - p_i)
> $
>
> 3. Gradient of SCE Loss:
> $
> \frac{\partial \mathcal{L}_{\text{SCE}}}{\partial z_i} = \alpha \cdot (p_i - y_i) + \beta \cdot \frac{y_i}{p_i} \cdot (1 - p_i)
> $
>
> For the true class ($i = y$):
> $
> \frac{\partial \mathcal{L}_{\text{SCE}}}{\partial z_y} = \alpha \cdot (p_y - 1) + \beta \cdot \frac{1}{p_y} \cdot (1 - p_y)
> $
>
> For all other classes ($i \neq y$):
> $
> \frac{\partial \mathcal{L}_{\text{SCE}}}{\partial z_i} = \alpha \cdot p_i + \beta \cdot \frac{0}{p_i} \cdot (1 - p_i) = \alpha \cdot p_i
> $
>
> We notice that, for both CE loss and SCE loss, their gradients reduce to 0 if and only if $p_i = y_i, \forall i$, which corresponds to the *overfitted case* analyzed in our paper. This implies that with sufficient model capacity and training (as is often the case with modern deep neural networks), the conclusions of our analysis remain valid even when robust loss functions are used.
>
>
> *We have also updated the manuscript accordingly with above new results in `Appendix F. Robust loss functions meet open-set noise`. If the reviewer has any other recommended baseline methods, we would be more than happy to further discuss and include them in the final version.*
>
>
> *[1] Zhang, Zhilu, and Mert Sabuncu. "Generalized cross entropy loss for training deep neural networks with noisy labels." Advances in neural information processing systems 31 (2018).*
>
> *[2] Wang, Yisen, et al. "Symmetric cross entropy for robust learning with noisy labels." Proceedings of the IEEE/CVF international conference on computer vision. 2019.*

---

> ### Author Response · Authors · 2024-11-23
> **Response to Reviewer op1H (Part 3/4)**
>
> > **W2: Comparison between different open-set noise with same pattern and noise ratio of close-set noise: *‘For Section 3.3.2, … how different open-set noise with same close-set label noise.’***
>
> Many thanks for the insightful question. Previously, we analyzed open-set noise by setting the closed-set noise to zero to simplify the analysis. We are pleased to confirm that our analysis can be extended to conditions with additional *'same pattern and noise ratio of closed-set noise'*.  Specifically, compared to the analysis in our paper, we no longer assume $T_{in} \neq \mathbf{I}$. Note that, similar to the main paper, we consider two proxy sample points $x_1$ and $x_2$ corresponding two different open-set noise modes. We also assume $[p^1_1, \dots,p^1_A] = [p^2_1, \dots,p^2_A]$ to focus solely on the impact of the noise modes.
>
> **1. Fitted case**:
>
> we first investigate the fitted case. Following the *error rate inflation* induced for fitted case (L260-L262) in the main paper, we have:
> - Easy open-set noise - $x_1$:
> $\Delta E_{x_1} = \max[p^1_1, \dots,p^1_A] - p_{\arg\max [\sum_{i=1}^{A+B}p^1_i T^1_{i1}, \dots , \sum_{i=1}^{A+B}p^1_i T^1_{iA}]} =  \max[p^1_1, \dots,p^1_A] - p_{\arg\max [\sum_{i=1}^{A} p^1_iT_{i1}+\frac{1}{A}\sum_{i=A+1}^{A+B} p^1_i, \dots, \sum_{i=1}^{A} p^1_iT_{iA}+\frac{1}{A}\sum_{i=A+1}^{A+B} p^1_i]}  = \max[p^1_1, \dots,p^1_A] - p_{\arg\max [\sum_{i=1}^{A} p^1_iT_{i1}, \dots, \sum_{i=1}^{A} p^1_iT_{iA}]}$
>
> - Hard open-set noise - $x_2$:
> $\Delta E_{x_2} =\max[p^2_1, ...,p^2_A] - p_{\arg\max [\sum_{i=1}^{A+B}p^2_i T^2_{i1}, ..., \sum_{i=1}^{A+B}p^2_i T^2_{iA}]} =\max[p^2_1, ...,p^2_A] - p_{\arg\max[\sum_{i=1}^{A} p^2_iT_{i1}+\sum_{b\in H_1}p^2_b,...,\sum_{i=1}^{A} p^2_iT_{iA}+\sum_{b\in H_A}p^2_b]}$.
>
>  Unfortunately, without extra assumptions on $T_{in}$ or $[p^1_1, \dots,p^1_A]$, to compare $\Delta E_{x_1}$ and $\Delta E_{x_2}$ is impossible. Here, we consider two conservative but realistic cases:
>
> **i. Concentration assumption of $[p^1_1, \dots,p^1_A]$**: in this case, we assume the probability $[p^1_1, \dots,p^1_A]$ concentrate on one specific class, say, $t$. We thus have $p^1_t \rightarrow 1, p^1_k \rightarrow 0, \forall k \neq t$.  In this case, we have:
> - Easy open-set noise - $x_1$:
> $\Delta E_{x_1} = \max[p^1_1, \dots,p^1_A] - p_{\arg\max [\sum_{i=1}^{A} p^1_iT_{i1}, \dots, \sum_{i=1}^{A} p^1_iT_{iA}]} \\
> \approx p^1_t - p_{\arg\max [p^1_tT_{t1}, \dots, p^1_tT_{tt}, \dots,p^1_tT_{tA}]}\\
>   \xrightarrow{\text{diagnomal-dominant noise transition matrix}} \\
> =0.$
>
> Note we normally implciitly assume a *diagnomal-dominant noise transition matrix*, that is, $\forall i, j\neq i, T_{ii} > T_{ij}$.
>
> - Hard open-set noise - $x_2$:
> $\Delta E_{x_2} =\max[p^2_1, ...,p^2_A] - p_{\arg\max[\sum_{i=1}^{A} p^2_iT_{i1}+\sum_{b\in H_1}p^2_b,...,\sum_{i=1}^{A} p^2_iT_{iA}+\sum_{b\in H_A}p^2_b]}\\
> \approx p^2_t - p_{\arg\max [p^2_tT_{t1}+\sum_{b\in H_1}p^2_b, \dots, p^2_tT_{tt}+\sum_{b\in H_t}p^2_b, \dots,p^2_tT_{tA}+\sum_{b\in H_A}p^2_b]}\\
> \geq 0.$
>
>
>
> **ii. Symmetric closed-set noise for $T_{in}$**: in this case, we assume a symmetric noise transition matrix $T$.
>
> - Easy open-set noise with $x_1$:
> $\Delta E_{x_1} = \max[p^1_1, \dots,p^1_A] - p_{\arg\max [\sum_{i=1}^{A} p^1_iT_{i1}, \dots, \sum_{i=1}^{A} p^1_iT_{iA}]} \\
> = \max[p^1_1, \dots,p^1_A] - p_{\arg\max [\sigma+ p^1_1T_{\Delta}, \dots, \sigma+ p^1_AT_{\Delta}]}\\
> =0.$
>
> - Hard open-set noise with $x_2$:
> $\Delta E_{x_2} =\max[p^2_1, ...,p^2_A] - p_{\arg\max [\sum_{i=1}^{A+B}p^2_i T^2_{i1}, ..., \sum_{i=1}^{A+B}p^2_i T^2_{iA}]} \\
> =\max[p^2_1, ...,p^2_A] - p_{\arg\max[\sum_{i=1}^{A} p^2_iT_{i1}+\sum_{b\in H_1}p^2_b,...,\sum_{i=1}^{A} p^2_iT_{iA}+\sum_{b\in H_A}p^2_b]}\\
> = p^2_t - p_{\arg\max [\sigma+ p^2_1T_{\Delta}+\sum_{b\in H_1}p^2_b,\dots,\sigma+ p^2_AT_{\Delta}+\sum_{b\in H_A}p^2_b]}\\
> \geq 0.$
>
> In above two cases, we have $\Delta E_{x_1} = 0$; thus we have $\Delta E_{x_1} \leq \Delta E_{x_2}$. **That is to say, under either of the two popular assumptions above, we arrive at the same conclusion: 'easy' open-set noise is less harmful than 'hard' open-set noise.**

---

> ### Author Response · Authors · 2024-11-23
> **Response to Reviewer op1H (Part 4/4)**
>
> **2. Overfitted case**:
>
> we then re-investigate the overfitted-case. Following the *error rate inflation* induced for overfitted case (L264-L267) in the main paper, we have:
> - Easy open-set noise with $x_1$:
> $\Delta E_{x_1} = \max[p^1_1, ...,p^1_A] - \sum_{i=1}^A (p^1_i\cdot \sum_{j=1}^{A+B}p^1_j T^1_{ji}) \\
> = \max[p^1_1, ...,p^1_A] - \sum_{i=1}^A \big(p^1_i\cdot (\sum_{j=1}^{A}p^1_j T^1_{ji} + \sum_{j=A+1}^{A+B}p^1_j T^1_{ji})\big)\\
>   \xrightarrow{T^1_{in}\neq \mathbf{I}, \ T^1_{out} = T^{easy}} \\
> = \max[p^1_1, ...,p^1_A] - \sum_{i=1}^A p^1_i (\sum_{j=1}^{A}p^1_j T^1_{ji}+\frac{1}{A}\sum_{i=A+1}^{A+B} p^1_i).$
>
> - Hard open-set noise with $x_2$:
> $\Delta E_{x_2} = \max[p^2_1, ...,p^2_A] - \sum_{i=1}^A (p^2_i\cdot \sum_{j=1}^{A+B}p^2_j T^2_{ji})\\
> = \max[p^2_1, ...,p^2_A] - \sum_{i=1}^A \big(p^2_i\cdot (\sum_{j=1}^{A}p^2_j T^2_{ji} + \sum_{j=A+1}^{A+B}p^2_j T^2_{ji})\big)\\
>   \xrightarrow{T^2_{in}\neq \mathbf{I}, \  T^2_{out} = T^{hard}} \\
> = \max[p^2_1, ...,p^2_A] - \sum_{i=1}^A p^2_i(\sum_{j=1}^{A}p^2_j T^2_{ji}+\sum_{j\in H_i}p^2_j)$
>
> Similarly, we have:
>
> $\Delta E_{x_1} - \Delta E_{x_2} = \sum_{i=1}^A p^2_i(\sum_{j=1}^{A}p^2_j T^2_{ji}+\sum_{j\in H_i}p^2_j)-\sum_{i=1}^A p^1_i (\sum_{j=1}^{A}p^1_j T^1_{ji}+\frac{1}{A}\sum_{i=A+1}^{A+B} p^1_i)
> =\sum_{i=1}^A p^1_i (\sum_{j\in H_i}p^1_j - \frac{1}{A}\sum_{i=A+1}^{A+B} p^1_i)$
>
> We note that the result aligns with L1052–L1056 in Appendix D.2. **Therefore, the presence of additional closed-set noise does not affect the conclusion in the overfitted case.**
>
> *We hope the above can further resolve the your concerns about. We have also updated the manuscript accordingly with above analysis in `Appendix D.3`.*
>
> > **W3: Experiments on more methods: *'Experiments on the main paper … add more LNL methods and discusses how different methods behave and align with the theorems in the main paper.'***
>
> Many thanks for the suggestion. We kindly refer you to our reply to W1.
>
> > ***Q1: 'For Figure 2 (c),(d), Authors state in the paper that "the presence of open-set noise degrades OOD detection performance, whereas, conversely, the presence of closed-set noise could even improve OOD detection performance.". However, from my observation, the closed-set noise does not improve detection performance from the Figure.'***
>
> We apologize for the misleading statement. Our intention was to convey that the results presented in Figures 2(c/d) demonstrate that, in the presence of open-set noise, the OOD detection performance is even better than the Clean Baseline (indicated by the dotted line in the figure). To better illustrate this intention, we have updated the manuscript in L483 - L487 and added the following clarification:
>
> *`For example, we notice that in the fitted case, the existence of open-set noise leads to steady improvement in OOD detection performance for both CIFAR100-O and ImageNet-O datasets, across different noise ratios.`*
>
> > ***Q2: 'In Section 3.3.2, Is it possible to assume the same pattern and noise ratio of close-set noise and then study how different open-set noise compare to each other?'***
>
> Many thanks for the question. We kindly refer you to our reply to W2.
>
> > ***Q3: 'For Figure 2, is it drawn when model converges for each case?'***
>
> Many thanks for the question. We apologize for a typo in the subcaptions of Figure 2 - the 'memorized case' in the old figure indeed refers to the *overfitted case*, where the model (PreResNet18) is trained for a sufficiently long time and converges. For the *fitted case*, we train a linear classifier on top of a pretrained frozen encoder, also, for a sufficiently long time. We believe this setup limits the model capacity while ensuring that the dataset is already well-represented, allowing us to approximate the *fitted case*.
>
> **Thank you once again for your time and valuable feedback, which have helped us improve our work. If there are any additional questions or clarifications needed, please do not hesitate to let us know. We would also kindly ask if you might consider revising your score, should our responses satisfactorily resolve the points you raised. We truly appreciate your consideration and look forward to your further comments.**

---

> > ### Comment · Reviewer_op1H · 2024-11-26
> >
> > I sincerely thank the authors for their detailed and thoughtful responses, especially for providing new experimental results and theoretical analyses to address my concerns. I understand the challenges of delivering these updates within the short rebuttal period, and I greatly appreciate their effort. Most of my concerns have now been satisfactorily addressed.
> >
> > Overall, while this paper leans more towards presenting an observation rather than introducing a novel method, I believe its findings offer valuable insights that could advance our understanding of open-set label noise in the learning with noisy labels (LNL) research domain. Therefore, I maintain my score, which leans towards acceptance.

---

> > > ### Author Response · Authors · 2024-11-30
> > >
> > > Dear Reviewer op1H,
> > >
> > > We sincerely apologize for reaching out to you again, and we hope this does not disrupt your Thanksgiving holiday if you celebrate it. **We deeply appreciate your acknowledgment of our rebuttal and your support for the acceptance of our paper. We sincerely wonder if you might consider increasing the rating and what further efforts we could make in this regard—your further support is crucial for this borderline submission.**
> > >
> > > Thank you once again for your time and support. We greatly appreciate your detailed review and valuable feedback on our submission. Your insights have been instrumental in helping us refine and improve the quality of our work. We are truly grateful for your assistance and look forward to your reply.
> > >
> > > Best regards,
> > >
> > > Authors of Submission 4732

---

> ### Author Response · Authors · 2024-11-26
> **Thanks to reviewer op1H for the recognition**
>
> Dear Reviewer op1H,
>
>
> Thank you for your kind response and recognition of our method. **In the context of the increasingly zero-sum dynamics of the ICLR community, your acknowledgement is particularly meaningful to us.** Your suggestion for robust loss functions and your insights on theoretical aspects are of great value in improving the quality of our paper.
>
> **We approach you with humility and sincerity to seek your guidance on whether there is any possibility for further adjustments that could earn your greater support.** We are more than willing to make further improvements and optimizations to the manuscript based on your suggestions to maximize its academic quality.
>
> Additionally, we would like to briefly mention that we have also explored **two potential methods for addressing open-set noise**, which are detailed in our responses to reviewer *rg5p* and in `Appendix G` of the updated manuscript. We hope this additional effort aligns with your expectations and earns your support.
>
> We would like to express our sincere gratitude for your continuous support and valuable feedback. We sincerely hope that this message will not cause you any inconvenience and look forward to your reply.
>
>
> With our deepest respect,
>
> Authors of submission 4732

---

### Author Response · Authors · 2024-11-23
**General response and summary of updated manuscript**

We sincerely thank the reviewers and the area chair for their time, effort, and constructive feedback on our manuscript. We appreciate the reviewer's recognition of the insights our work brings to the LNL community, especially in the study of open-set noise. **We carefully reviewed and addressed all the comments and have made corresponding revisions to address the concerns raised. The key updates to our manuscript include**:

- Minor Revisions in Main Paper:
  1. Fixed typos and grammatical errors.
  2. Condensed the contributions section.
  3. Updated the presentation in the experiments `Section 4.1` for improved clarity and consistency.
- Additional Analysis in the Appendix:
  1. Expanded experiments on the performance of robust loss functions under open-set noise in `Appendix F`.
  2. Added comparisons of 'hard' vs. 'easy' open-set noise under scenarios with additional same and fixed closed-set noise in `Appendix D.3`.
  3. Conducted preliminary explorations on identifying and learning effectively with 'hard' open-set noise in `Appendix G`.

We hope these updates address the reviewers' concerns and strengthen the overall quality of the paper. We are grateful for the opportunity to further refine our work and look forward to additional feedback.

---

### Public Comment · ~Jiawei_Ge2 · 2024-12-05
**Enquiry for further insights**

Excellent work! It’s a really interesting and attractive idea to combine the OOD detection task with the original classification task. However, I’m wondering if the authors could provide further insights into the relationship between classification accuracy and OOD detection performance. For example, could optimizing one objective negatively impact the other?

---

### Meta-Review · Area_Chair_z4MB · 2024-12-17

**Metareview:**

This submission received the ratings of three reviewers, which recommended 6, 3 and 5, averaging 4.67. Given the plenty of competitive submissions in ICLR, this stands at a score below the borderline. The AC has noticed that a reviewer who rated 3 provided a very short review, which has been ignored in evaluating the overall contribution. One reviewer rating 5 mainly concerned about the contribution given the progress of the whole area and previous similar constructions, including the potential solution, which is well addressed based on the author rebuttal.  To some extent, I agree with the reviewer rating 6's comment about presenting an observation rather than introducing a novel method, however, given the competitive submission, I also feel that there is still some effort that can be done to improve the submission, like some potential solutions to be explored along with the analysis and observation, which will make the submission outstanding in the peer submissions. Currently, I regret to tend to recommend rejection considering the overall review of two constructive reviewers, and hope the comments help the improvement of the submission.

The AC.

**Additional Comments On Reviewer Discussion:**

1. Only presenting the observation and analysis mentioned by one constructive reviewer.

Not well addressed.

---

### Decision · Program_Chairs · 2025-01-22

Reject